# Selective Disclosure Watermarking for Large Language Models

Xuyang Chen [1]  Xiang Li [1]  Yangxinyu Xie [1]  Qi Long [1]

## Abstract

Watermarking methods embed imperceptible and verifiable signals into text generated by large language models (LLMs). Existing approaches include zero-bit schemes for distinguishing synthetic text from human writing and multi-bit schemes for embedding metadata. However, current multi-bit watermarking methods do not allow selective disclosure: verifying any part of the watermark requires revealing the entire embedded message. This lack of control leads to unnecessary information exposure and raises privacy concerns. We propose **Hie**rarchical Vocabulary **Ro**uting (HeRo), a watermarking framework that enables selective disclosure of embedded metadata. The method recursively partitions the vocabulary and distributes watermark information across hierarchical layers, so that different verifiers can decode only the portions of the payload corresponding to their access level. We show that the proposed scheme preserves the unbiasedness of the underlying sampling process and thus maintains text quality. Experiments demonstrate that our framework supports fine-grained access control while achieving high detection accuracy and low latency. Code is available at https://github.com/xuyangc03/hero-watermark.

## 1. Introduction

The widespread adoption of large language models (LLMs) has fundamentally transformed content creation across many domains. Their ability to produce high-quality, human-like text at scale brings substantial benefits, but also introduces serious social risks (Bommasani et al., 2021; Weidinger et al., 2021), including academic dishonesty, harmful content generation, and misinformation (Ranade

et al., 2021; Huang & Sun, 2023). Early attempts to distinguish AI-generated text from human writing relied on post-hoc, content-based detectors (Mitchell et al., 2023). However, these methods (e.g., GPTZero, OpenAI's Detector) have shown limited effectiveness as LLMs continue to improve (Weber-Wulff et al., 2023). At the same time, regulators are increasingly emphasizing provider-side content authentication mechanisms, such as transparency and machine-readable marking requirements in the EU AI Act (The European Parliament and the Council of the European Union, 2024; Laux et al., 2024) and emerging state-level regulations in California (California State Legislature, 2024).

Watermarking has emerged as a promising alternative by embedding algorithmically verifiable yet human-imperceptible signals directly into model outputs. A large body of prior work on LLM watermarking (Kirchenbauer et al., 2023; Aaronson & Kirchner, 2024; Zhao et al., 2024; Dathathri et al., 2024) focuses on zero-bit schemes, whose goal is to distinguish LLM-generated text from human writing. While effective for detection, it is essentially a binary classification problem, which is insufficient for many real-world provenance tracking and auditing applications.

To address this limitation, multi-bit watermarking methods (Fernandez et al., 2023; Wang et al., 2024; Yoo et al., 2024; Jiang et al., 2025; Feng et al., 2025) allow providers to embed richer metadata into generated text, such as model versions, timestamps, or user and session identifiers. However, existing designs typically do not support partial verification: any party with access to the verification key can recover the entire embedded payload. In practice, this all-or-nothing disclosure creates a fundamental deployment challenge. Many real systems require two conflicting properties: (1) broad verifiability, so that the public can confirm coarse provenance information, and (2) restricted auditing, so that sensitive metadata can be accessed only by authorized parties. For example, a platform moderator may only need to verify that content was produced by a specific provider or model family, whereas a privileged auditor may need access to confidential fields such as internal identifiers or user information (The European Parliament and the Council of the European Union, 2024; California State Legislature, 2024). Without fine-grained access control, providers must choose between full opacity and complete transparency, limiting the practical applicability of existing watermarking schemes.

[1]University of Pennsylvania. Correspondence to: Xuyang Chen <xuyangc@sas.upenn.edu>.

*Proceedings of the 43rd International Conference on Machine Learning*, Seoul, South Korea. PMLR 306, 2026. Copyright 2026 by the author(s).

In general, an effective multi-bit watermarking system should satisfy several key requirements. It should preserve generation quality, remain robust to common text perturbations, incur minimal decoding latency, and scale efficiently to large online corpora. Beyond these basic properties, we argue that it should also provide sufficient capacity to encode rich metadata and, crucially, enable selective disclosure: a verifier should recover only the information corresponding to its authorization level, without learning any additional embedded data.

To this end, we introduce a selective-disclosure watermarking framework based on hierarchical vocabulary partitioning. The framework organizes the vocabulary into a nested tree structure and composes a base watermarking rule recursively. At the top level, the vocabulary is partitioned to embed a coarse message, and the selected subset is then further partitioned to embed deeper payload layers (see Figure 1 for an illustration). This recursive construction creates a dependency chain in which lower-level payloads are statistically concealed within higher layers. As a result, the framework naturally supports hierarchical verification: a standard verifier can decode only the root-level signal (e.g., detection or coarse provenance), while deeper layers appear as noise, whereas privileged verifiers can resolve nested partitions to recover sensitive metadata. We instantiate the framework using Gumbel-based watermarking and evaluate detection accuracy, text quality, robustness under representative perturbations, and computational cost.

Our main contributions are as follows:

- We propose a framework for selective-disclosure multi-bit watermarking in LLM decoding, enabling hierarchical verification under different authorization levels.

- We provide theoretical guarantees on statistical unbiasedness and formalize selective disclosure by showing that unauthorized parties cannot recover payloads beyond random guessing.

- We provide an efficient, batching-friendly GPU implementation and extensive experiments demonstrating strong detectability, text quality preservation, robustness to perturbations, and substantially lower generation/decoding latency than publicly released multi-bit watermarking implementations.

## 2. Related Work

LLM watermarking methods can be categorized by when the watermark is embedded. Post-processing watermarking rewrites already-generated text to inject detectable patterns, e.g., via controlled lexical substitutions such as synonym replacement (Yang et al., 2023; Munyer et al., 2024). In this work we focus on inference-time watermarking, which embeds signals by modifying the sampling procedure during generation to correlate token selection with a secret key.

**Zero-bit Watermarking.** A representative zero-bit watermark is the Green-Red List scheme (Kirchenbauer et al., 2023), which partitions the vocabulary into green and red lists and biases sampling toward green tokens. Detection is performed by testing whether the generated text contains an unusually high fraction of green tokens. Though simple, this watermark distorts the model's output distribution and can degrade generation quality. A subsequent line of work aims to debias this watermark through reweighting (Hu et al., 2024; Wu et al., 2024; Xie et al., 2025). In parallel, Aaronson & Kirchner (2024); Kuditipudi et al. (2024); Dathathri et al. (2024) propose statistically unbiased sampling procedures that provably preserve the model output distribution while still enabling reliable detection. Robust watermarking designs are further studied in (Kuditipudi et al., 2024; Zhao et al., 2024; Li et al., 2026; Qu et al., 2025).

**Multi-bit Watermarking.** Multi-bit watermarking schemes embed metadata into generated text to meet auditing needs that go beyond binary detection. Many multi-bit methods build on the Green-Red List paradigm (Kirchenbauer et al., 2023) by making the vocabulary partition message-dependent, so that different metadata payloads correspond to different preferred token subsets. Early designs (Fernandez et al., 2023; Qu et al., 2025) construct message-specific green lists by cyclically shifting a keyed vocabulary permutation according to the message. Wang et al. (2024) uses a proxy language model to form higher-quality vocabulary partitions, improving text quality at the cost of additional computation and an extra modeling assumption. MPAC (Yoo et al., 2024) introduces a position allocation technique. Instead of encoding the entire payload at every token, it embeds a message subunit at each position so that different parts of the payload accumulate evidence across the text. This position-allocation viewpoint has been widely adopted by subsequent multi-bit methods. StealthInk (Jiang et al., 2025) and BiMark (Feng et al., 2025) design reweighting mechanisms to achieve statistical unbiasedness, aiming to preserve text quality while retaining watermarking capacity.

**Metadata Access Control and Selective Disclosure.** Despite progress on generation quality, robustness, and payload capacity, granular access control in LLM watermarking has received comparatively little attention. Most existing schemes implicitly assume an all-or-nothing credential model: a verifier either can decode the watermark and recover the full payload or cannot decode anything at all. The closest related direction is designated-detector watermarking (Huang et al., 2024), where cryptographic techniques restrict who can detect the presence of a watermark. Unlike

designated-detector watermarking, which controls who can detect a watermark, our goal is selective disclosure: different keys reveal different subsets of the embedded metadata from the same text. To our knowledge, no existing LLM watermarking method provides role-based partial verification of embedded metadata within the same text. Our work formalizes this selective-disclosure requirement and proposes a mechanism that enables hierarchical verification: low-privilege verifiers can validate or decode only coarse information, while deeper payload layers remain statistically concealed without the corresponding authorization.

## 3. Preliminaries

**Language Modeling Fundamentals.** A language model $\mathcal{M}$ is an autoregressive probabilistic model defined over a discrete vocabulary $\mathcal{V}$ of size $V = |\mathcal{V}|$. At each generation step, the model predicts the next token based on all preceding tokens. Given a prefix sequence $x_{<t} = (x_1, \ldots, x_{t-1})$, the model produces a logit vector $l_t \in \mathbb{R}^V$ at step $t$. The next-token prediction (NTP) distribution $P_{\mathcal{M}}(\cdot \mid x_{<t})$ is obtained by applying the softmax function to these logits:

$$P_{\mathcal{M}}(x_t = v \mid x_{<t}) = \frac{\exp((l_t)_v)}{\sum_{j=1}^{V} \exp((l_t)_j)}. \qquad (1)$$

In standard generation, the next token $x_t$ is sampled from this NTP distribution: $x_t \sim P_{\mathcal{M}}(\cdot \mid x_{<t})$.

**Watermarking via Modified Sampling.** Watermarking embeds a discrete payload $m$ (or a metadata message) into generated text by modifying only the token sampling process, while keeping the underlying NTP distribution unchanged. At each generation step $t$, a pseudo-random function (PRF) $\mathcal{A}$ (Goldreich et al., 1986) maps a local context window of size $h$, $x_{t-h:t-1} = (x_{t-h}, \ldots, x_{t-1})$, together with a secret key $\xi$, to a pseudorandom variable $\zeta_t = \mathcal{A}(x_{t-h:t-1}, \xi) \in \mathbb{R}^V$. The next token is generated by a (deterministic) sampling function $\mathcal{S}$:

$$x_t = \mathcal{S}(P_{\mathcal{M}}(\cdot \mid x_{<t}), \zeta_t). \qquad (2)$$

**Multi-bit Payloads and Position Allocation.** In multi-bit watermarking, the payload $m$ is represented as a sequence of message segments $m = (m_1, \ldots, m_K)$. Following the position-allocation strategy of MPAC (Yoo et al., 2024), a rule $p(t) \in \{1, \ldots, K\}$ specifies which segment is embedded at generation step $t$, so that token $x_t$ encodes only the single segment $m_{p(t)}$ rather than the full payload $m$. For example, if the total payload has 24 bits and each allocated position carries 2 bits, then the payload is divided into $K = 12$ segments, and the position-allocation rule assigns these segments to generation steps. During detection, the detector applies a function $\hat{m} = \mathcal{D}(x, \xi)$ to recover the embedded payload from a generated $x$.

**Definition 3.1** (Statistical Unbiasedness). A watermarking scheme is statistically unbiased if, conditioned on any prefix $x_{<t}$, the marginal distribution of each generated token $v \in \mathcal{V}$ equals the original NTP distribution:

$$\mathbb{P}_{\zeta_t}(\mathcal{S}(P_{\mathcal{M}}(\cdot \mid x_{<t}), \zeta_t, m) = v) = P_{\mathcal{M}}(x_t = v \mid x_{<t}),$$

where the probability is taken over the randomness induced by the secret key.

**Statistical Unbiasedness.** We require the watermarking scheme to be statistically unbiased (see Definition 3.1), so that watermarking preserves the model's NTP distribution and does not degrade generation quality. A canonical example of an unbiased sampling rule is Gumbel-Max sampling in Definition 3.2. We use it as a convenient sampling rule to illustrate our framework.

**Definition 3.2** (Gumbel-Max Sampling). The pseudorandom variable $\zeta_t$ is defined as $\zeta_t = (U_{t,v})_{v \in \mathcal{V}}$, where $U_{t,v} \overset{i.i.d.}{\sim} \text{Uniform}(0, 1)$. This construction assigns independent uniform randomness to each candidate token. The next token is selected as $x_t = \arg\max_{v \in \mathcal{V}} (l_t)_v - \log(-\log(U_{t,v}))$.

## 4. Methodology

**Overview.** In this section, we introduce our watermarking method. The key idea is to embed information by guiding the token sampling process through a hierarchical partition of the vocabulary. Starting from the full vocabulary, we progressively refine the partition and select a smaller subset of candidate tokens at each stage. The choice of which subset to refine is controlled by pseudorandomness derived from a secret key and is tied to a specific segment of the payload. By repeating this process across multiple layers, the method embeds a multi-bit message while preserving the original sampling distribution.

**A Two-Layer Example.** Figure 1 illustrates the procedure using a two-layer construction. Consider a two-level payload $(m^{(1)}, m^{(2)})$. At the first layer, the vocabulary is partitioned into $K_1 = 4$ disjoint chunks, each assigned an aggregated probability equal to the sum of the original next-token probabilities of the tokens it contains. Equivalently, each chunk can be viewed as a "meta-token," and a sampling step is performed over these chunks according to their aggregated probabilities. The pseudorandomness used in this step is obtained from the entry corresponding to $m^{(1)}$ in the first key table $\boldsymbol{\xi}^{(1)}$, i.e., $\boldsymbol{\xi}^{(1)}(m^{(1)})$. Once a chunk is selected, the same procedure is applied recursively within the selected chunk to embed $m^{(2)}$: the chunk is further partitioned, aggregated probabilities are computed, and sampling is performed again using pseudorandomness $\boldsymbol{\xi}^{(2)}(m^{(2)})$ determined by the next message segment. Because each selection step samples according to the appropriate (aggregated

*Figure 1.* Hierarchical vocabulary routing on a toy vocabulary of size $V = 16$ with a chunking schedule $K = (K_1) = (4)$. At stage $\ell = 1$, the current candidate set (blue) is partitioned into $K_1 = 4$ contiguous chunks (dotted connectors). Using the level-1 key table $\xi^{(1)}$ and message $m^{(1)}$, the sampler draws a chunk (solid arrows) and restricts the candidate set to the selected chunk (green). After the routing stage, the final token (black) is sampled from the remaining candidates using key table $\xi^{(2)}$ and message $m^{(2)}$.

or conditional) probability distribution, the overall procedure remains statistically unbiased, and the final sampled token follows the original NTP distribution.

### 4.1. General $L$-Layer Generation

Now, we describe how to embed an $L$-level payload into generated text using hierarchical vocabulary routing. The routing structure is specified by a chunking schedule $K = (K_1, \ldots, K_{L-1})$, where $K_\ell$ denotes the number of partitions (chunks) at routing stage $\ell$. The sampler starts from the full vocabulary $\mathcal{C}^{(0)} = \mathcal{V}$ and proceeds through $L$ sequential stages, repeatedly refining the candidate set until a single token is selected. Earlier stages make coarse routing decisions over large vocabulary regions, while later stages refine the decision within the selected region; each routing decision carries one level of payload information. The full procedure is summarized in Algorithm 1.

**Stages** $\ell = 1, \ldots, L - 1$ **(Chunk Routing).** At routing stage $\ell$, the current candidate chunk $\mathcal{C}^{(\ell-1)}$ is partitioned into $K_\ell$ contiguous chunks (line 3 in Algorithm 1)

$$(\mathcal{C}_1^{(\ell)}, \mathcal{C}_2^{(\ell)}, \ldots, \mathcal{C}_{K_\ell}^{(\ell)}) = \text{Partition}(\mathcal{C}^{(\ell-1)}),$$

with sizes differing by at most one.[1] We then coarse-grain the token distribution by aggregating probabilities within each chunk, which yields a categorical distribution over the $K_\ell$ chunks (line 4–5). Using the pseudorandom variable $\zeta^{(\ell)}$ associated with the payload component $m^{(\ell)}$, we apply the sampling function $\mathcal{S}$ to select a chunk $\mathcal{C}^{(\ell)} = \mathcal{C}_{s_\ell}^{(\ell)}$ and restrict the candidate set to the selected chunk (line 8).

**Stage** $\ell = L$ **(Final Token Sampling).** After $L - 1$ routing decisions, we obtain a final candidate chunk $\mathcal{C}^{(L-1)}$. We treat each token in this chunk as a separate category and sample the final token $x_t$ from the newly normalized distribution restricted to this subset, guided by the last-level payload $m^{(L)}$. Each generated token induces a nested path of vocabulary subsets: $\mathcal{V} = \mathcal{C}^{(0)} \supset \mathcal{C}^{(1)} \supset \cdots \supset \mathcal{C}^{(L-1)} \supset \mathcal{C}^{(L)} = \{x_t\}$, where the routing decision at stage $\ell$ is controlled by the level-$\ell$ message component. The following theorem formalizes that this hierarchical sampling procedure preserves the original NTP distribution.

**Theorem 4.1** (Statistical Unbiasedness). *The hierarchical routing sampler is statistically unbiased: the final sampled token $x_t$ follows the original NTP distribution.*

### 4.2. Decoding Multi-Level Payload

---

[1]Suppose $v, k \in \mathbb{N}^+$ and $v = ak + r$, where $0 \leq r < k$, we have $v = a(k - r) + (a + 1)r$. This means we can always partition a large chunk into small chunks with max difference size one: $k - r$ chunks with size $a$ and $r$ chunks with size $a + 1$.

**Algorithm 1** Generation via HeRo at step $t$

**Require:** NTP distribution $P_t(\cdot) = P_{\mathcal{M}}(\cdot \mid x_{<t})$, context window $x_{(t-h):(t-1)}$, PRF $\mathcal{A}$, sampling function $\mathcal{S}$, payload $m = (m^{(1)}, \ldots, m^{(L)})$, key tables $\{\boldsymbol{\xi}^{(\ell)}\}_{\ell=1}^L$, chunking schedule $(K_1, \ldots, K_{L-1})$

**Ensure:** Sampled token $x_t$

1: Offset $o \leftarrow 0$ and current chunk size $v \leftarrow V$
2: **for** $\ell = 1$ **to** $L - 1$ **do**
3:     Partition $[o, o + v) = \cup_{i=1}^{K_\ell} \mathcal{C}_i^{(\ell)}$ into $K_\ell$ contiguous chunks
4:     $w_i^{(\ell)} \leftarrow \sum_{x \in \mathcal{C}_i^{(\ell)}} P_t(x)$ for $i = 1, \ldots, K_\ell$
5:     Normalize $(w_1^{(\ell)}, \ldots, w_{K_\ell}^{(\ell)})$ to obtain a probability distribution over the $K_\ell$ chunks
6:     Identify the secret key $\xi \leftarrow \boldsymbol{\xi}^{(\ell)}(m^{(\ell)})$
7:     Get pseudorandom $\zeta_t^{(\ell)} \leftarrow \mathcal{A}(x_{(t-h):(t-1)}, \xi)$
8:     Sample the next chunk $s_{t,\ell} \leftarrow \mathcal{S}((w_i^{(\ell)})_{i=1}^{K_\ell}, \zeta_t^{(\ell)})$
9:     Update offset $o$ and chunk size $v$ according to the selected chunk $\mathcal{C}_{s_{t,\ell}}^{(\ell)}$
10: **end for**
11: Identify the secret key $\xi \leftarrow \boldsymbol{\xi}^{(L)}(m^{(L)})$
12: Get pseudorandom $\zeta_t^{(L)} \leftarrow \mathcal{A}(x_{(t-h):(t-1)}, \xi)$
13: Sample index $s_{t,L} \leftarrow \mathcal{S}(P_t|_{[o,o+v)}, \zeta_t^{(L)})$
14: **return** $x_t \leftarrow o + s_{t,L}$

**Algorithm 2** Message Decoding via HeRo

**Require:** Generated sequence $x = (x_1, \ldots, x_T)$, vocabulary size $V$, context window size $h$, PRF $\mathcal{A}$, evidence function Ev, key tables $\{\boldsymbol{\xi}^{(\ell)}\}_{\ell=1}^L$, chunking schedule $(K_1, \ldots, K_{L-1})$

**Ensure:** Decoded payload $(\hat{m}^{(1)}, \ldots, \hat{m}^{(L)})$

1: {Stage 1: per-token evidence}
2: **for** $t = 1$ **to** $T$ **do**
3:     Recover the chunk-index path $(s_{t,1}, \ldots, s_{t,L})$ from $x_t$
4:     **for** $\ell = 1$ **to** $L$ **do**
5:       **for** $a = 0$ **to** $2^{b_\ell} - 1$ **do**
6:         $\zeta_t^{(\ell)}(a) \leftarrow \mathcal{A}\Big(x_{(t-h):(t-1)}, \boldsymbol{\xi}^{(\ell)}(a)\Big)$
7:         $E_t^{(\ell)}(a) \leftarrow \text{Ev}\Big(\zeta_t^{(\ell)}(a), s_{t,\ell}\Big)$
8:       **end for**
9:     **end for**
10: **end for**
11: {Stage 2: aggregate evidence and decode each level}
12: **for** $\ell = 1$ **to** $L$ **do**
13:     Decode $\hat{m}^{(\ell)}$ from the per-token evidences $\{E_t^{(\ell)}(a)\}_{t,a}$
14: **end for**
15: **return** $(\hat{m}^{(1)}, \ldots, \hat{m}^{(L)})$

The decoding procedure follows the reverse logic of generation (Algorithm 2). Given a generated sequence $x = (x_1, \ldots, x_T)$, the decoder processes each token position independently and infers, from the token identity alone, the hierarchical routing decisions made during generation. Concretely, the deterministic partition rule maps each token $x_t$ to a unique sequence of routing indices $\{s_{t,\ell}\}_{\ell=1}^L$. For $\ell = 1, \ldots, L-1$, the index $s_{t,\ell} \in \{1, \ldots, K_\ell\}$ records the selected chunk at routing stage $\ell$. At the final stage, we equivalently treat each token in the last selected chunk as a singleton chunk, so $s_{t,L}$ again denotes a chunk index. This sequence serves as the observable footprint of the hierarchical routing process.

We now describe the decoding procedure more formally. Fix a level $\ell$, a party with access to the level-$\ell$ key table $\boldsymbol{\xi}^{(\ell)}$ computes the pseudorandom variable for each candidate message value $a \in \{0, \ldots, 2^{b_\ell} - 1\}$ as

$$\zeta_t^{(\ell)}(a) = \mathcal{A}\Big(x_{(t-h):(t-1)}, \boldsymbol{\xi}^{(\ell)}(a)\Big).$$

The decoder then combines $\zeta_t^{(\ell)}(a)$ with the observed chunk index $s_{t,\ell}$ to form a per-token evidence value

$$E_t^{(\ell)}(a) = \text{Ev}\Big(\zeta_t^{(\ell)}(a), s_{t,\ell}\Big).$$

The evidence function Ev is designed so that for an incorrect candidate $a \neq m^{(\ell)}$, $E_t^{(\ell)}(a)$ follows a known null

distribution, while the true message produces stochastically larger evidence. Aggregating the evidence across token positions (e.g., by summation) yields a score for each candidate message, and the decoded payload at level $\ell$ is given by

$$\hat{m}^{(\ell)} = \arg \max_{a \in \{0, \ldots, 2^{b_\ell} - 1\}} \text{Agg}(\{E_t^{(\ell)}(a)\}).$$

**Example 1.** *As a concrete example, we use Gumbel-Max sampling as the sampling rule $\mathcal{S}$ in this work. For routing stages $\ell < L$, the pseudorandom variable $\zeta_t^{(\ell)}(a) = (U_{t,i})_{i=1}^{K_\ell}$ consists of $K_\ell$ i.i.d. $\text{Uniform}(0,1)$ random variables, which are used to sample the next chunk from the partition $(\mathcal{C}_1^{(\ell)}, \mathcal{C}_2^{(\ell)}, \ldots, \mathcal{C}_{K_\ell}^{(\ell)})$. At the final stage $\ell = L$, the same construction applies to the last selected chunk. The evidence function is the Aaronson score used for detection (Aaronson & Kirchner, 2024):*

$$\text{Ev}(\zeta, s) = -\log(1 - \zeta(s)),$$

*where $\zeta(s)$ denotes the $s$-th entry of the vector $\zeta$. Finally, we simply set $\text{Agg}(\{E_t^{(\ell)}(a)\}) = \sum_t E_t^{(\ell)}(a)$.*

Without access to the level-$\ell$ key table $\boldsymbol{\xi}^{(\ell)}$, a verifier cannot reproduce the pseudorandom variables corresponding to the true message value $m^{(\ell)}$. As a result, the per-token evidence $\{E_t^{(\ell)}(a)\}_a$ is statistically indistinguishable across candidate messages, and decoding at level $\ell$ reduces to random

guessing. The following theorem formalizes this selective disclosure property.

**Theorem 4.2** (Selective Disclosure). *Fix a disclosure level $k \in \{1, \ldots, L\}$. Consider a verifier that possesses only the key tables $\boldsymbol{\xi}^{(1)}, \ldots, \boldsymbol{\xi}^{(k)}$. Then, for any $\ell > k$, the verifier cannot decode $m^{(\ell)}$ beyond random guessing.*

### 4.3. Computational Complexity Analysis

We analyze the computational cost of generation and decoding for a single generation step $t$. Throughout, we assume that Gumbel-Max sampling is used at all routing stages.

**Generation.** At step $t$, each routing stage $\ell \in \{1, \ldots, L-1\}$ partitions the current candidate set into $K_\ell$ contiguous chunks and computes their probability masses $\{p_i\}_{i=1}^{K_\ell}$. This can be implemented efficiently by first computing the prefix-sum (CDF) $F_t(u) = \sum_{v \leq u} P_t(v)$ in $O(V)$ time. Each chunk mass can then be obtained using two CDF queries, yielding an $O(K_\ell)$ cost per stage. Under Gumbel-Max sampling, drawing a chunk index from a $K_\ell$-way categorical distribution also takes $O(K_\ell)$ time.

After $L-1$ routing stages, the remaining candidate set has size approximately $V / \prod_{\ell=1}^{L-1} K_\ell$, and sampling the final token costs $O\left(V / \prod_{\ell=1}^{L-1} K_\ell\right)$. Overall, the per-step generation complexity is $O\left(V + \sum_{\ell=1}^{L-1} K_\ell + V / \prod_{\ell=1}^{L-1} K_\ell\right) = O(V)$. In practice, we implement generation in a batched manner to improve wall-clock throughput. The additional overhead introduced by watermarking is negligible compared to the model forward pass, since our complexity depends only on the vocabulary size, which remains essentially constant across scales within each model family (see Table 4 in Appendix D.1).

**Decoding.** Decoding operates solely on the generated text and the key tables. At each level $\ell$, the decoder evaluates $2^{b_\ell}$ candidate message values. Under Gumbel-Max sampling, the evidence function $\mathrm{Ev}(\zeta, s)$ can be computed without reconstructing the full pseudorandom vector $\zeta$, as it requires only the random variate $\zeta_s$ associated with the observed token index $s$. By using a counter-based pseudorandom number generator (Salmon et al., 2011), this computation takes $O(1)$ time and $O(1)$ memory per candidate. As a result, the total per-step decoding complexity is $O\left(\sum_{\ell=1}^{L} 2^{b_\ell}\right)$.

## 5. Experiments

**Considered Watermarks.** We evaluate the Hierarchical Vocabulary Routing framework instantiated with Gumbel-Max sampling. We consider both single-level and two-level hierarchical payload configurations in the main text. The two-level configuration enables selective disclosure by separating public verification (first level) from private auditing (second level) using different secret keys. The single-level configuration does not provide hierarchical access control and therefore serves as an internal baseline. We additionally compare against prior multi-bit watermarking methods, including MPAC (Yoo et al., 2024), StealthInk (Jiang et al., 2025), and BiMark (Feng et al., 2025). We further investigate deeper hierarchies (up to 8 levels) and report the results in Appendix E.3.

**Models and Datasets.** We conduct experiments using `Llama2-7B` (Touvron et al., 2023). For text generation, we use the `C4 realnewslike` dataset (Raffel et al., 2020), which contains formal journalistic content, and the `OpenGen` dataset (Krishna et al., 2023), which covers conversational and creative text. From each dataset, we sample 1,000 documents and truncate a fixed number of initial tokens to form generation prompts. We report results on `C4` in the main text, with additional results on `OpenGen` provided in Appendix E.1 and Appendix E.2.

**Evaluation Metrics.** We evaluate watermarking schemes along five dimensions. **Selective disclosure** is assessed by comparing decoding outcomes across authorization levels. **Detectability** is measured by bit-level message decoding accuracy. **Text quality** is quantified using perplexity (PPL). **Efficiency** is reported as wall-clock generation and decoding time. **Robustness** is evaluated by decoding accuracy under random replacement and roundtrip translation.

### 5.1. Selective Disclosure Evaluation

To evaluate selective disclosure, we consider two-level hierarchical payloads, where the first level is treated as public and the second as private. This setting induces two verifier capabilities: (i) **Full authorization**: the verifier holds secret keys for both levels and can decode both payloads and (ii) **Public-only authorization**: the verifier holds only the public secret key and can decode the public payload, while private payload decoding is no better than random guessing.

We fix the total payload size at 24 bits and vary the per-level allocation as $(b_1, b_2) \in \{(1,1), (2,1), (2,2)\}$. Figure 2 illustrates a clear trade-off between public and private decoding accuracy for the fully authorized verifier. As the number of public chunks $K_1$ increases, more signal is allocated to the public level, leading to improved and eventually saturated public decoding accuracy. Conversely, private decoding accuracy decreases as less signal remains available for the private level. This trade-off is best balanced at $K_1 = 20$, which we adopt in subsequent experiments.

Finally, we verify the selective-disclosure guarantee by showing that a public-only verifier achieves approximately chance-level accuracy ($\approx 50\%$) on the private payload across all settings (see dotted line in Figure 2), consistent

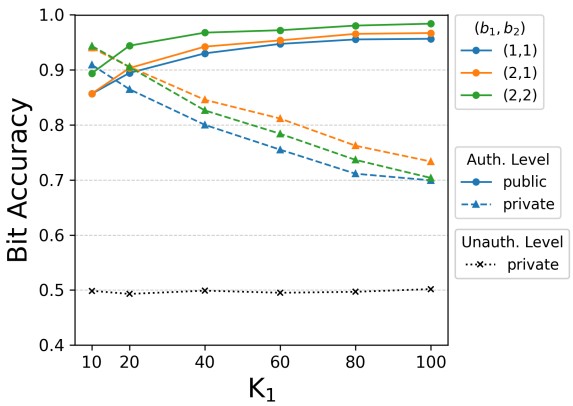

*Figure 2.* Bit accuracy as a function of the number of first-layer chunks $K_1$: public payload accuracy (solid), private payload accuracy for fully authorized verifiers (dashed) and private payload accuracy under public-only authorization (dotted).

with Theorem 4.2.

### 5.2. Detectability and Text Quality

**Detection Accuracy.** We evaluate detection accuracy across watermarking methods. Our method is denoted by HeRo, with $HeRo^1(b_1)$ and $HeRo^2(b_1, b_2)$ representing single-level and two-level configurations, respectively.

Table 1 reports bit-level decoding accuracy under a fixed generation budget of 200 tokens for total payload sizes $m \in \{12, 24, 36, 48\}$. Across all payload sizes, our method consistently achieves stronger detectability than MPAC, StealthInk, and BiMark. In particular, the single-level configuration $HeRo^1$ achieves near-perfect decoding at moderate payload sizes (24 and 36 bits) and remains highly accurate even at 48 bits. The two-level configuration $HeRo^2$ exhibits a modest accuracy reduction relative to $HeRo^1$, while still matching or outperforming prior multi-bit baselines at the same total payload size. Importantly, $HeRo^2$ additionally supports selective disclosure (Section 5.1) while maintaining strong detectability.

Figure 3 further shows that decoding accuracy improves with longer generations for all methods, and that our method achieves higher accuracy with fewer generated tokens.

**Text Quality Preservation.** Figure 4 visualizes the PPL distributions. The distribution under our method closely overlaps with the unwatermarked baseline, whereas MPAC exhibits a noticeable shift toward higher perplexity values, implying a large quality degradation.

### 5.3. Computational Efficiency

We measure wall-clock latency for both watermark embedding during generation and message decoding. We bench-

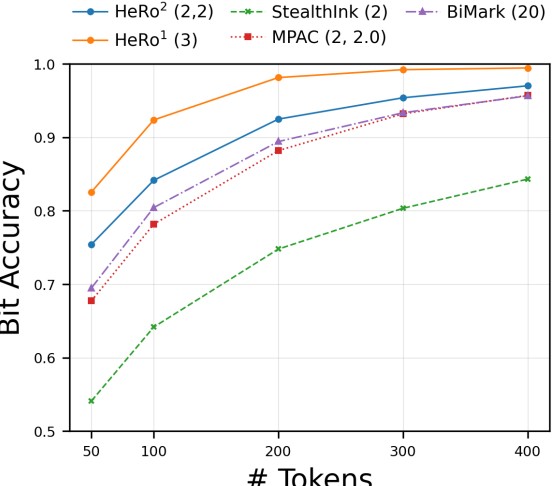

*Figure 3.* Comparison of bit accuracy for multi-bit watermarking methods as a function of token budget when embedding a 24-bit payload.

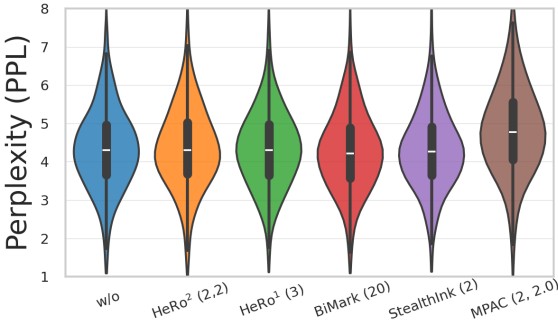

*Figure 4.* Violin plots of perplexity (PPL) for `Llama2-7B` generations on `C4`, comparing multi-bit watermarking methods with the unwatermarked baseline (w/o). Results are computed over 200 generated tokens.

mark our implementation against publicly released codebases for MPAC and BiMark.

**Generation Latency.** Real-world LLM services typically process multiple requests concurrently and rely on batching to improve GPU utilization and amortize overhead. We generate 1,000 continuations with a 24-bit embedded payload, and vary the generation batch size in $\{1, 4, 16, 32, 64\}$. For each method, we report per-token generation latency, computed as total elapsed generation time divided by the number of generated tokens.

Figure 5 summarizes the results of generation latency. The top panel reports absolute per-token latency (ms/token), while the bottom panel shows relative overhead compared to the baseline. Our implementation naturally supports batched inference and incurs only a small and stable overhead (approximately 4–5%) over the non-watermarked base-

*Table 1.* Comparison of bit accuracy (B.Acc.↑) and median perplexity (PPL) for `Llama2-7B` generations on `C4` across payload sizes (12, 24, 36, and 48 bits). Results are computed over 200 generated tokens. MPAC is parameterized by $(b, \delta)$, where $\delta$ denotes the bias added to green-token logits; StealthInk by $(b)$; BiMark by $(d)$, where $d$ is the number of probability reweighting steps; and our method by $\text{HeRo}^L(b_1, \ldots, b_L)$ with $b_\ell$ bits of payload allocated at level $\ell$.

| Watermark | S.D. | 12 Bits | | 24 Bits | | 36 Bits | | 48 Bits | |
|---|---|---|---|---|---|---|---|---|---|
| | | B.Acc.↑ | PPL | B.Acc.↑ | PPL | B.Acc.↑ | PPL | B.Acc.↑ | PPL |
| w/o | — | — | 4.31 | — | 4.31 | — | 4.31 | — | 4.31 |
| MPAC (1, 2.0) | — | 0.954 | 4.69 | 0.876 | 4.71 | 0.840 | 4.69 | 0.800 | 4.64 |
| MPAC (2, 2.0) | — | 0.959 | 4.78 | 0.882 | 4.76 | 0.835 | 4.75 | 0.802 | 4.75 |
| StealthInk (1) | — | 0.934 | 4.36 | 0.845 | 4.33 | 0.805 | 4.32 | 0.769 | 4.32 |
| StealthInk (2) | — | 0.843 | 4.27 | 0.748 | 4.31 | 0.731 | 4.33 | 0.687 | 4.37 |
| BiMark (5) | — | 0.866 | 4.31 | 0.767 | 4.28 | 0.731 | 4.26 | 0.702 | 4.33 |
| BiMark (10) | — | 0.914 | 4.25 | 0.827 | 4.32 | 0.788 | 4.30 | 0.746 | 4.31 |
| BiMark (20) | — | 0.961 | 4.21 | 0.894 | 4.32 | 0.852 | 4.29 | 0.812 | 4.26 |
| $\text{HeRo}^1$ (2) | — | 0.993 | 4.32 | 0.966 | 4.28 | 0.933 | 4.31 | 0.901 | 4.33 |
| $\text{HeRo}^1$ (3) | — | **0.997** | 4.31 | **0.981** | 4.33 | **0.953** | 4.29 | **0.919** | 4.32 |
| $\text{HeRo}^2$ (1,1) | ✓ | 0.950 | 4.31 | 0.880 | 4.31 | 0.831 | 4.32 | 0.798 | 4.33 |
| $\text{HeRo}^2$ (2,1) | ✓ | 0.970 | 4.30 | 0.904 | 4.28 | 0.856 | 4.27 | 0.823 | 4.32 |
| $\text{HeRo}^2$ (2,2) | ✓ | **0.975** | 4.30 | **0.925** | 4.36 | **0.873** | 4.30 | **0.833** | 4.28 |

line across all batch sizes, with a mild decreasing trend as batch size increases. In contrast, MPAC and BiMark exhibit substantial overhead across all batch sizes, and their overhead does not diminish with batching, reflecting limited batching support in the released implementations.

*Table 2.* Decoding latency ($\mu$s/token) using `Llama2-7B` tokenizer. Values are averaged over approximately 400,000 tokens.

| Watermark | Batch Size | | | | |
|---|---|---|---|---|---|
| | 1 | 4 | 16 | 32 | 64 |
| MPAC (2, 2.0) | 488.13 | 479.30 | 491.43 | 489.15 | 480.43 |
| BiMark (20) | 130.43 | 130.22 | 130.68 | 131.63 | 133.23 |
| $\text{HeRo}^1$ (3) | **7.75** | **2.43** | **1.66** | **1.57** | **1.44** |
| $\text{HeRo}^2$ (2,2) | **10.34** | **3.08** | **1.85** | **1.59** | **1.48** |

**Message Decoding Latency.** Fast decoding is important for large-scale auditing settings, where messages must be recovered from large text corpora. During message decoding, the embedded payload $m$ is recovered from the generated text. We measure wall-clock decoding time and report average latency in $\mu$s/token over approximately 400,000 tokens using `Llama2-7B` tokenizer. Table 2 shows decoding latency as a function of the number of texts processed per batch.

Our decoding procedure is compatible with standard batched execution and achieves substantially lower decoding latency than the baselines, with additional speedups as batch size increases. In contrast, the publicly released implementations of MPAC and BiMark perform decoding on a per-text basis

and are CPU-based. We thus invoke their decoding procedures independently for each example in the batch, which does not yield meaningful speedups as batch size increases.

### 5.4. Robustness Analysis

We evaluate robustness under a wide range of perturbation strategies. For random replacement (REP), random insertion (INS), and random deletion (DEL), a fraction $p \in \{0.05, 0.1, 0.2\}$ of tokens are randomly substituted, inserted, or deleted, respectively, where larger values correspond to more aggressive corruption. Roundtrip translation (RT) translates the watermarked text from English to French and back using OPUS-MT models (Tiedemann & Thottingal, 2020). We also evaluate paraphrasing attacks via DIPPER (Krishna et al., 2023) under three configurations of lexical diversity and order diversity: $(20, 0)$, $(0, 20)$, $(20, 20)$, where the two parameters control the degree of lexical substitution and sentence reordering, respectively.

Table 3 reports bit accuracy under perturbation strategies for a 24-bit payload. For random replacement (REP), insertion (INS), and deletion (DEL), decoding accuracy consistently decreases as the corruption rate $p$ increases. Roundtrip translation (RT) produces degradation comparable to the strongest random perturbation setting ($p = 0.2$). Under DIPPER paraphrasing, accuracy degrades as lexical and order diversity increases. Across all perturbation settings, our framework consistently achieves the highest decoding accuracy among the evaluated methods, indicating stronger preservation of recoverable watermark signals under text corruption.

*Table 3.* Bit accuracy (B.Acc.↑) under attacks on `C4` using `Llama2-7B` with a 24-bit payload and 200 generated tokens. REP: random replacement; INS: random insertion; DEL: random deletion; RT: roundtrip translation; DIPPER: paraphrasing attacks with controllable lexical and order diversity.

| Watermark | REP | | | INS | | | DEL | | | RT | DIPPER | | |
|---|---|---|---|---|---|---|---|---|---|---|---|---|---|
| | 0.05 | 0.1 | 0.2 | 0.05 | 0.1 | 0.2 | 0.05 | 0.1 | 0.2 | | (20,0) | (0,20) | (20,20) |
| MPAC (2, 2.0) | 0.809 | 0.739 | 0.622 | 0.814 | 0.754 | 0.648 | 0.827 | 0.766 | 0.663 | 0.650 | 0.669 | 0.794 | 0.646 |
| StealthInk (2) | 0.677 | 0.620 | 0.552 | 0.684 | 0.630 | 0.559 | 0.690 | 0.643 | 0.575 | 0.569 | 0.582 | 0.665 | 0.573 |
| BiMark (20) | 0.833 | 0.768 | 0.658 | 0.838 | 0.778 | 0.677 | 0.845 | 0.793 | 0.692 | 0.687 | 0.703 | 0.817 | 0.691 |
| HeRo[1] (3) | **0.949** | **0.898** | **0.752** | **0.952** | **0.906** | **0.782** | **0.958** | **0.920** | **0.798** | **0.780** | **0.800** | **0.931** | **0.778** |
| HeRo[2] (2,2) | **0.871** | **0.800** | **0.677** | **0.868** | **0.809** | **0.707** | **0.879** | **0.825** | **0.710** | **0.707** | **0.723** | **0.848** | **0.701** |

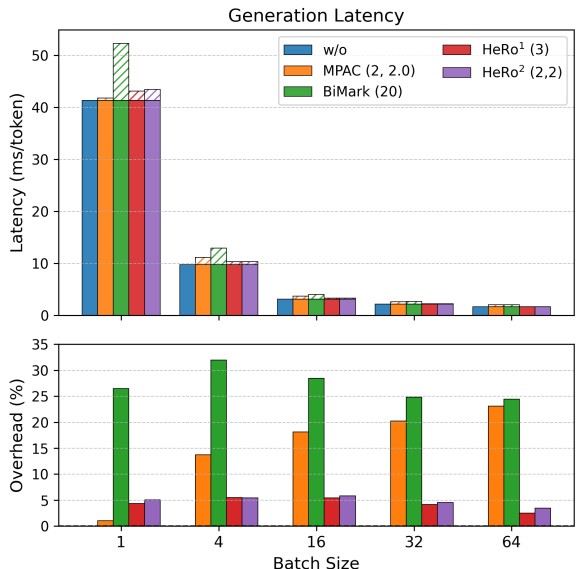

*Figure 5.* Generation latency under batched inference for `Llama2-7B` generations on `C4` with a 24-bit payload. **Top:** per-token generation latency (ms/token). **Bottom:** relative overhead over the non-watermarked baseline (w/o). Batch size denotes the number of prompts processed concurrently.

## 6. Discussion

This work introduces a hierarchical watermarking framework for selective metadata disclosure in LLM-generated text. We provide theoretical guarantees for selective disclosure and evaluate the framework under both single-level and multi-level payload configurations. Our experiments demonstrate strong selective disclosure performance, robustness to common text perturbations, and favorable detectability-quality trade-offs. Moreover, under a fixed generation budget, we observe an inherent trade-off between selective disclosure and detectability. Formally characterizing this trade-off and identifying its fundamental limits remain open problems, which we leave for future work.

## Acknowledgements

We thank Weijie Su for helpful discussions and valuable feedback on this work.

## Impact Statement

This paper proposes a hierarchical multi-bit watermarking framework that enables selective disclosure of embedded metadata in LLM-generated text. The intended positive impact is to support scalable provenance verification and auditing while reducing unnecessary information exposure: different verifiers can decode only the portions of the payload corresponding to their access level, which can improve privacy compared to all-or-nothing multi-bit watermarking.

Like other watermarking techniques, our approach may also introduce risks. If deployed without appropriate governance, embedded metadata could be used to track users and even create chilling effects on speech. In addition, watermark detectors are inherently statistical and may yield false positives or false negatives. Therefore, watermark evidence should not be treated as sole proof of authorship or user attribution in high-stakes decisions.

We believe these risks can be mitigated by (i) using selective disclosure to minimize the default exposed payload, (ii) employing strong key management and audit logging for privileged decoding, and (iii) treating watermark verification as one signal among others within a transparent policy. Overall, we view selective disclosure as a step toward more privacy-preserving and accountable deployment of watermarking for LLM-generated text.

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

# A. Proof of Theorem 4.1

We prove that at any generation step $t$, the hierarchical routing sampler preserves the original next-token probability (NTP) distribution $P_t$ over the full vocabulary $\mathcal{V}$. The hierarchical sampler first routes through a sequence of nested partitions and then samples a token from the final selected chunk.

**Hierarchical chunks.** Fix a step $t$ and let $P_t(\cdot) = P_{\mathcal{M}}(\cdot \mid x_{<t})$ be the model's NTP distribution on $\mathcal{V}$, so $\sum_{x \in \mathcal{V}} P_t(x) = 1$. The sampler begins with the full vocabulary $\mathcal{C}^{(0)} = \mathcal{V}$. For each routing stage $\ell \in \{1, \ldots, L-1\}$, the current candidate set $\mathcal{C}^{(\ell-1)}$ is partitioned into $K_\ell$ contiguous disjoint chunks by a deterministic rule:

$$(\mathcal{C}_1^{(\ell)}, \ldots, \mathcal{C}_{K_\ell}^{(\ell)}) = \text{Partition}(\mathcal{C}^{(\ell-1)}),$$

satisfying

$$\bigcup_{i=1}^{K_\ell} \mathcal{C}_i^{(\ell)} = \mathcal{C}^{(\ell-1)} \text{ and } \mathcal{C}_i^{(\ell)} \cap \mathcal{C}_j^{(\ell)} = \emptyset \ (i \neq j).$$

The aggregated probability mass of chunk $i$ at stage $\ell$ is

$$w_i^{(\ell)} = \sum_{x \in \mathcal{C}_i^{(\ell)}} P_t(x).$$

**Chunk selection.** At routing stage $\ell$, the sampler treats the $K_\ell$ chunks $\mathcal{C}_1^{(\ell)}, \ldots, \mathcal{C}_{K_\ell}^{(\ell)}$ as meta-tokens and draws an index $s_{t,\ell} \in \{1, \ldots, K_\ell\}$ according to

$$\mathbb{P}(s_{t,\ell} = i \mid \mathcal{C}^{(\ell-1)}) = \frac{w_i^{(\ell)}}{\sum_{x \in \mathcal{C}^{(\ell-1)}} P_t(x)} = \frac{\sum_{x \in \mathcal{C}_i^{(\ell)}} P_t(x)}{\sum_{x \in \mathcal{C}^{(\ell-1)}} P_t(x)}.$$

The candidate set is then restricted to the selected chunk:

$$\mathcal{C}^{(\ell)} = \mathcal{C}_{s_{t,\ell}}^{(\ell)}.$$

**Final token selection.** After $L-1$ routing stages, the sampler holds a final chunk $\mathcal{C}^{(L-1)}$ and draws the token $x_t$ from $P_t$ restricted to this set:

$$\mathbb{P}(x_t = v \mid \mathcal{C}^{(L-1)}) = \frac{P_t(v)}{\sum_{x \in \mathcal{C}^{(L-1)}} P_t(x)} \quad \text{for } v \in \mathcal{C}^{(L-1)}.$$

We now show the marginal distribution of the output satisfies $\mathbb{P}(x_t = v) = P_t(v)$ for every token $v \in \mathcal{V}$.

*Proof of Theorem 4.1.* Fix a token $v \in \mathcal{V}$. Because the partition rule is deterministic, $v$ determines a unique nested chunk path through the hierarchy. For each $\ell \in \{1, \ldots, L-1\}$, let $i_\ell(v) \in \{1, \ldots, K_\ell\}$ be the unique index with

$$v \in \mathcal{C}_{i_\ell(v)}^{(\ell)}$$

and write

$$\mathcal{C}^{(\ell)} := \mathcal{C}_{i_\ell(v)}^{(\ell)}$$

for the unique chunk containing $v$ at stage $\ell$. Define the routing event

$$R_v := \bigcap_{\ell=1}^{L-1} \{s_{t,\ell} = i_\ell(v)\}.$$

Sampling $v$ requires the sampler to follow this path, so $\{x_t = v\} \subseteq R_v$. Therefore,

$$\mathbb{P}(x_t = v) = \mathbb{P}(x_t = v \mid R_v)\mathbb{P}(R_v).$$

**Routing factor.** At stage $\ell$, the chunk containing $v$ has aggregated mass

$$w_{i_\ell(v)}^{(\ell)} = \sum_{x \in \mathcal{C}^{(\ell)}} P_t(x).$$

Thus

$$
\begin{aligned}
\mathbb{P}(R_v) &= \prod_{\ell=1}^{L-1} \mathbb{P}\left(s_{t,\ell} = i_\ell(v) \,\middle|\, \bigcap_{j=1}^{\ell-1}\{s_{t,j} = i_j(v)\}\right) \\
&= \prod_{\ell=1}^{L-1} \frac{w_{i_\ell(v)}^{(\ell)}}{\sum_{x \in \mathcal{C}^{(\ell-1)}} P_t(x)} \\
&= \prod_{\ell=1}^{L-1} \frac{\sum_{x \in \mathcal{C}^{(\ell)}} P_t(x)}{\sum_{x \in \mathcal{C}^{(\ell-1)}} P_t(x)} \\
&= \sum_{x \in \mathcal{C}^{(L-1)}} P_t(x),
\end{aligned}
$$

where the last equality follows from telescoping together with $\mathcal{C}^{(0)} = \mathcal{V}$ and $\sum_{x \in \mathcal{V}} P_t(x) = 1$.

**Final-stage factor.** Conditioned on $R_v$, the final candidate set is $\mathcal{C}^{(L-1)}$, so

$$\mathbb{P}(x_t = v \mid R_v) = \mathbb{P}(x_t = v \mid \mathcal{C}^{(L-1)}) = \frac{P_t(v)}{\sum_{x \in \mathcal{C}^{(L-1)}} P_t(x)}.$$

Multiplying the two factors, we obtain

$$\mathbb{P}(x_t = v) = P_t(v).$$

Because $v$ is arbitrary, we prove the final marginal of $x_t$ matches the original NTP distribution.

$\square$

## B. Proof of Theorem 4.2

Fix a level $\ell > k$, a verifier without access to $\boldsymbol{\xi}^{(\ell)}$ can only observe the generated sequence $x$ and the authorized key tables $\boldsymbol{\xi}^{(1)}, \ldots, \boldsymbol{\xi}^{(k)}$. The verifier has no statistical way to distinguish which candidate is used at generation time, as formalized by the following assumption.

**Assumption B.1.** Conditioned on the observed sequence $x$ and the authorized tables $\boldsymbol{\xi}^{(1)}, \ldots, \boldsymbol{\xi}^{(k)}$, the random vector $\left(T^{(\ell)}(0), \ldots, T^{(\ell)}(2^{b_\ell} - 1)\right)$ is exchangeable with respect to permutations of the candidate index set $\{0, \ldots, 2^{b_\ell} - 1\}$. Equivalently, for any permutation $\pi$ of $\{0, \ldots, 2^{b_\ell} - 1\}$,

$$\left(T^{(\ell)}(0), \ldots, T^{(\ell)}(2^{b_\ell} - 1)\right) \stackrel{d}{=} \left(T^{(\ell)}(\pi(0)), \ldots, T^{(\ell)}(\pi(2^{b_\ell} - 1))\right).$$

We now prove that exchangeability implies chance-level decoding at the bit level. For a $b_\ell$-bit message $a \in \{0, \ldots, 2^{b_\ell} - 1\}$, we let $b_r(a)$ denote the $r$-th bit of $a$. Let $\hat{m}^{(\ell)}$ be any decoder output based on $(x, \boldsymbol{\xi}^{(1)}, \ldots, \boldsymbol{\xi}^{(k)})$, and $\hat{b}_r = b_r(\hat{m}^{(\ell)})$.

**Lemma B.2.** *Assume $m^{(\ell)}$ is uniform on $\{0, \ldots, 2^{b_\ell} - 1\}$. Under Assumption B.1, for any decoder,*

$$\mathbb{P}\left[\hat{b}_r = b_r(m^{(\ell)})\right] = \tfrac{1}{2}, \quad \forall r \in \{1, \ldots, b_\ell\}.$$

*Proof.* Fix $r \in \{1, \ldots, b_\ell\}$. Let $S_0 = \{a : b_r(a) = 0\}$ and $S_1 = \{a : b_r(a) = 1\}$, we have $|S_0| = |S_1| = 2^{b_\ell - 1}$. Because $m^{(\ell)}$ is uniform over $\{0, \ldots, 2^{b_\ell} - 1\}$, $b_r(m^{(\ell)})$ is uniform on $\{0, 1\}$. Consider the permutation $\pi_r(a)$ that flips the $r$-th bit of $a$, i.e., $b_r(\pi_r(a)) = 1 - b_r(a)$ for all $a$. By Assumption B.1, we have

$$\left(T^{(\ell)}(0), \ldots, T^{(\ell)}(2^{b_\ell} - 1)\right) \stackrel{d}{=} \left(T^{(\ell)}(\pi_r(0)), \ldots, T^{(\ell)}(\pi_r(2^{b_\ell} - 1))\right).$$

Applying any decoder rule $\phi(\cdot)$ to the above expression, it follows that $\hat{m}^{(\ell)} = \phi(T^{(\ell)}(0), \ldots, T^{(\ell)}(2^{b_\ell} - 1))$ has the same distribution as $\phi(T^{(\ell)}(\pi_r(0)), \ldots, T^{(\ell)}(\pi_r(2^{b_\ell} - 1)))$. Thus $b_r(\hat{m}^{(\ell)})$ is equal in distribution to $1 - b_r(\hat{m}^{(\ell)})$, we have $\mathbb{P}[\hat{b}_r = 0] = \mathbb{P}[\hat{b}_r = 1] = 1/2$. Because $b_r(m^{(\ell)})$ is uniform, we prove

$$\mathbb{P}[\hat{b}_r = b_r(m^{(\ell)})] = \tfrac{1}{2}.$$

$\square$

*Proof of Theorem 4.2.* For any $\ell > k$, by applying Lemma B.2 to each bit position $r = 1, \ldots, b_\ell$, we prove the chance-level decoding accuracy of unauthorized verifiers. $\square$

## C. Detailed Algorithms

We present the algorithms instantiated with Gumbel-Max sampling. Algorithm 3 expands Algorithm 1 to show the full generation pipeline, and Algorithm 4 expands Algorithm 2 to show the full decoding pipeline.

## D. Experiment Details

All experiments are run on the NVIDIA A40 GPU with 48 GB memory. For model generation configuration, we set the temperature to $1.0$, while keeping other decoding hyperparameters (e.g., top-$p$ and top-$k$) as the model defaults. We use a context window of size $h = 4$ to derive the watermark seed from the recent $h$ tokens and a secret key.

The seed used to generate pseudorandom variables is determined by the key and the current $h$-gram context. Repetition is commonly observed in neural text generation (Fu et al., 2021; Xu et al., 2022). If the model revisits an $h$-gram that has appeared earlier in the same generation, watermarking may reuse the same seed and produce repetition loops. To mitigate this issue, we apply a context masking strategy during generation, i.e., we skip watermarking at the positions where the current $h$-gram has occurred previously. This avoids embedding at positions that would introduce redundant and correlated signals, which may hurt rather than help detection. Context masking therefore removes low-quality embedding positions rather than reducing useful signals. Empirically, detectability remains high under this strategy, consistent with similar approaches adopted in prior watermarking work (Hu et al., 2024; Jiang et al., 2025).

**Row-Wise Seeded Uniform RNG.** Watermarking requires generating pseudorandom variables with context-dependent seeds, which vary across sequences and time steps in a batch. To support efficient batched inference, we implement a CUDA kernel that generates uniform random variates with row-wise seeds using cuRAND's Philox state. For efficient decoding, we further provide an indexed variant that returns a single uniform variate per row at a specified vocabulary index, avoiding generating the full vocabulary-length vector when only one entry is needed.

### D.1. Model Size vs. Vocabulary Size

Table 4 reports parameter counts, model vocabulary sizes, and release dates for representative LLM families: OPT (Zhang et al., 2022), Llama 2 (Touvron et al., 2023), Qwen2.5 (Yang et al., 2024), and gpt-oss (Agarwal et al., 2025). Within each model family, the vocabulary size stays nearly constant across model scales, while the number of parameters grows by orders of magnitude.

### D.2. Batching Strategy

In our computational efficiency experiments (Section 5.3), we measure latency under static batching, where all sequences in a batch are padded to the same length and processed synchronously. We choose this setting to ensure a fair comparison against open-source baselines, which are typically implemented using standard deep learning frameworks without specialized inference optimizations. We acknowledge that modern LLM serving systems often employ continuous batching (e.g., vLLM (Kwon et al., 2023) and Orca (Yu et al., 2022)) to reduce padding overhead and improve throughput. An optimized integration with continuous batching is not explored here and we leave it as future work.

| Family | Model | Release date | Params | Vocab size |
|--------|-------|--------------|--------|------------|
| OPT | OPT-125M | 2022-05-03 | 125M | 50,272 |
| OPT | OPT-350M | 2022-05-03 | 350M | 50,272 |
| OPT | OPT-1.3B | 2022-05-03 | 1.3B | 50,272 |
| OPT | OPT-2.7B | 2022-05-03 | 2.7B | 50,272 |
| OPT | OPT-6.7B | 2022-05-03 | 6.7B | 50,272 |
| OPT | OPT-13B | 2022-05-03 | 13B | 50,272 |
| OPT | OPT-30B | 2022-05-03 | 30B | 50,272 |
| OPT | OPT-66B | 2022-05-03 | 66B | 50,272 |
| OPT | OPT-175B | 2022-05-03 | 175B | 50,272 |
| Llama 2 | Llama-2-7B | 2023-07-18 | 7B | 32,000 |
| Llama 2 | Llama-2-13B | 2023-07-18 | 13B | 32,000 |
| Llama 2 | Llama-2-70B | 2023-07-18 | 70B | 32,000 |
| Qwen2.5 | Qwen2.5-0.5B | 2024-09-19 | 0.5B | 151,936 |
| Qwen2.5 | Qwen2.5-1.5B | 2024-09-19 | 1.5B | 151,936 |
| Qwen2.5 | Qwen2.5-3B | 2024-09-19 | 3B | 151,936 |
| Qwen2.5 | Qwen2.5-7B | 2024-09-19 | 7B | 152,064 |
| Qwen2.5 | Qwen2.5-14B | 2024-09-19 | 14B | 152,064 |
| Qwen2.5 | Qwen2.5-32B | 2024-09-19 | 32B | 152,064 |
| Qwen2.5 | Qwen2.5-72B | 2024-09-19 | 72B | 152,064 |
| gpt-oss | gpt-oss-20b | 2025-08-05 | 21B | 201,088 |
| gpt-oss | gpt-oss-120b | 2025-08-05 | 117B | 201,088 |

*Table 4.* Representative LLM families at different parameter scales, with tokenizer vocabulary sizes and release dates. Parameter counts grow by orders of magnitude, while vocabulary size remains nearly constant within each family.

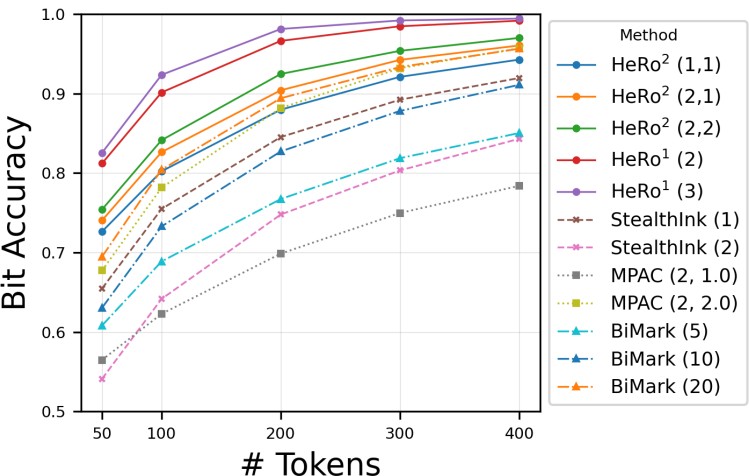

*Figure 6.* Comparison of bit accuracy for multi-bit watermarking methods as a function of token budget when embedding a 24-bit payload.

*Table 5.* TPR (↑) at target false positive rates ($10^{-2}, 5 \cdot 10^{-3}, 10^{-3}, 5 \cdot 10^{-4}$) for `Llama2-7B` generations on `C4` using HeRo$^2$ (2,2) across payload sizes (12, 24, 36, and 48 bits). Results are computed over 200 generated tokens.

| FPR | 12 Bits | 24 Bits | 36 Bits | 48 Bits |
|-----|---------|---------|---------|---------|
| $10^{-2}$ | 0.997 | 0.995 | 0.997 | 0.995 |
| $5 \cdot 10^{-3}$ | 0.996 | 0.995 | 0.997 | 0.993 |
| $10^{-3}$ | 0.989 | 0.988 | 0.985 | 0.972 |
| $5 \cdot 10^{-4}$ | 0.982 | 0.981 | 0.984 | 0.961 |

# E. Full Experiment Results

## E.1. Detectability

This section provides the full detectability results to complement the main text analysis in Section 5.2. Table 5 reports the true positive rate (TPR) achieved by HeRo$^2$ (2,2) for `Llama2-7B` generations on `C4` across payload sizes (12, 24, 36, and 48 bits) at several target false positive rates (FPR). Our method achieves strong detection performance under stringent FPR constraints, which suggests reliable identification with 200 tokens.

Tables 6–8 summarize bit-level decoding accuracy for `Llama2-7B` on both `C4` and `OpenGen` datasets, under generation length of 200 and 400 tokens, and across total payload sizes $m \in \{12, 24, 36, 48\}$. Across settings, the same trends observed in the main text persist: HeRo$^1$ attains strong (often near saturated) decoding accuracy at moderate payload sizes, while HeRo$^2$ exhibits a modest reduction relative to HeRo$^1$ yet remains competitive with or better than prior multi-bit baselines at the same total payload size. Figure 6 plots decoding accuracy as a function of the generation length for a 24-bit payload. In addition to the configurations shown in the main paper, we include a broader set of hyperparameter settings for a more comprehensive comparison.

*Table 6.* Comparison of bit accuracy (B.Acc.↑) and median perplexity (PPL) for `Llama2-7B` generations on `C4` across payload sizes (12, 24, 36, and 48 bits). Results are computed over 400 generated tokens.

| Watermark | S.D. | 12 Bits | | 24 Bits | | 36 Bits | | 48 Bits | |
|---|---|---|---|---|---|---|---|---|---|
| | | B.Acc.↑ | PPL | B.Acc.↑ | PPL | B.Acc.↑ | PPL | B.Acc.↑ | PPL |
| w/o | — | — | 4.16 | — | 4.16 | — | 4.16 | — | 4.16 |
| MPAC (1, 2.0) | — | 0.987 | 4.55 | 0.950 | 4.57 | 0.917 | 4.56 | 0.882 | 4.49 |
| MPAC (2, 2.0) | — | 0.987 | 4.65 | 0.957 | 4.65 | 0.920 | 4.62 | 0.886 | 4.59 |
| StealthInk (1) | — | 0.976 | 4.13 | 0.920 | 4.13 | 0.882 | 4.10 | 0.845 | 4.12 |
| StealthInk (2) | — | 0.925 | 4.05 | 0.843 | 4.12 | 0.817 | 4.15 | 0.767 | 4.18 |
| BiMark (5) | — | 0.934 | 4.14 | 0.850 | 4.12 | 0.807 | 4.12 | 0.773 | 4.17 |
| BiMark (10) | — | 0.970 | 4.08 | 0.911 | 4.13 | 0.866 | 4.10 | 0.829 | 4.10 |
| BiMark (20) | — | 0.989 | 4.03 | 0.956 | 4.05 | 0.923 | 4.08 | 0.892 | 4.08 |
| HeRo$^1$ (2) | — | 0.998 | 4.14 | 0.992 | 4.13 | 0.980 | 4.13 | 0.963 | 4.12 |
| HeRo$^1$ (3) | — | **0.998** | 4.10 | **0.994** | 4.10 | **0.986** | 4.14 | **0.974** | 4.14 |
| HeRo$^2$ (1,1) | ✓ | 0.982 | 4.11 | 0.943 | 4.13 | 0.904 | 4.13 | 0.869 | 4.11 |
| HeRo$^2$ (2,1) | ✓ | 0.989 | 4.13 | 0.960 | 4.14 | 0.931 | 4.13 | 0.900 | 4.14 |
| HeRo$^2$ (2,2) | ✓ | **0.991** | 4.16 | **0.970** | 4.17 | **0.940** | 4.10 | **0.912** | 4.10 |

## E.2. Perplexity

We provide the complete perplexity (PPL) results corresponding to Section 5.2. Specifically, we report the distribution of PPL for `Llama2-7B` generations on `C4` and `OpenGen` datasets, varying the watermark payload size in $\{12, 24, 36, 48\}$ bits and the evaluation length in $\{200, 400\}$ generated tokens. For each configuration, we compare the unwatermarked baseline (w/o) against a comprehensive set of multi-bit watermarking methods, including additional hyperparameter settings beyond those presented in the main paper. Figures 7–22 summarize the results using violin plots.

## E.3. Study of Hierarchy Depth

To study how hierarchy depth affects detection performance and generation quality, we evaluate bit accuracy and perplexity under four configurations: HeRo$^1$(8), HeRo$^2$(4,4), HeRo$^4$(2,2,2,2), and HeRo$^8$(1,1,1,1,1,1,1,1). Here, HeRo$^L$($b_1, \cdots, b_L$) denotes an $L$-level hierarchy in which the $\ell$-th layer discloses $b_\ell$ bits.

As shown in Table 9, detection generally becomes more difficult as the hierarchy becomes deeper. We observe that the performance gap across settings becomes smaller as the token budget increases. More broadly, the performance depends jointly on the hierarchy depth and the partition structure across levels, and there remains room to further optimize deeper configurations. Table 10 reports perplexity across all settings. The PPL remains very close to that of text generated without

*Table 7.* Comparison of bit accuracy (B.Acc.↑) and median perplexity (PPL) for `Llama2-7B` generations on `OpenGen` across payload sizes (12, 24, 36, and 48 bits). Results are computed over 200 generated tokens.

| Watermark | S.D. | 12 Bits | | 24 Bits | | 36 Bits | | 48 Bits | |
|---|---|---|---|---|---|---|---|---|---|
| | | B.Acc.↑ | PPL | B.Acc.↑ | PPL | B.Acc.↑ | PPL | B.Acc.↑ | PPL |
| w/o | — | — | 3.96 | — | 3.96 | — | 3.96 | — | 3.96 |
| MPAC (1, 2.0) | — | 0.944 | 4.40 | 0.861 | 4.40 | 0.823 | 4.42 | 0.781 | 4.31 |
| MPAC (2, 2.0) | — | 0.948 | 4.43 | 0.859 | 4.37 | 0.819 | 4.39 | 0.782 | 4.39 |
| StealthInk (1) | — | 0.912 | 3.86 | 0.820 | 3.91 | 0.787 | 3.94 | 0.747 | 3.94 |
| StealthInk (2) | — | 0.812 | 3.84 | 0.728 | 3.95 | 0.716 | 3.97 | 0.671 | 4.00 |
| BiMark (5) | — | 0.853 | 3.92 | 0.751 | 3.90 | 0.721 | 3.89 | 0.694 | 3.98 |
| BiMark (10) | — | 0.898 | 3.91 | 0.800 | 3.90 | 0.767 | 3.89 | 0.733 | 3.85 |
| BiMark (20) | — | 0.944 | 3.85 | 0.875 | 3.83 | 0.832 | 3.91 | 0.798 | 3.90 |
| HeRo[1] (2) | — | 0.986 | 3.88 | 0.949 | 3.92 | 0.906 | 3.96 | 0.878 | 3.93 |
| HeRo[1] (3) | — | **0.991** | 3.94 | **0.967** | 3.99 | **0.933** | 3.97 | **0.899** | 3.90 |
| HeRo[2] (1,1) | ✓ | 0.929 | 3.91 | 0.858 | 3.89 | 0.813 | 3.92 | 0.778 | 3.94 |
| HeRo[2] (2,1) | ✓ | 0.952 | 3.96 | 0.887 | 3.92 | 0.834 | 3.88 | 0.799 | 3.89 |
| HeRo[2] (2,2) | ✓ | **0.962** | 3.95 | **0.898** | 3.86 | **0.850** | 3.99 | **0.810** | 3.92 |

*Table 8.* Comparison of bit accuracy (B.Acc.↑) and median perplexity (PPL) for `Llama2-7B` generations on `OpenGen` across payload sizes (12, 24, 36, and 48 bits). Results are computed over 400 generated tokens.

| Watermark | S.D. | 12 Bits | | 24 Bits | | 36 Bits | | 48 Bits | |
|---|---|---|---|---|---|---|---|---|---|
| | | B.Acc.↑ | PPL | B.Acc.↑ | PPL | B.Acc.↑ | PPL | B.Acc.↑ | PPL |
| w/o | — | — | 3.70 | — | 3.70 | — | 3.70 | — | 3.70 |
| MPAC (1, 2.0) | — | 0.977 | 4.08 | 0.930 | 4.14 | 0.891 | 4.14 | 0.855 | 4.04 |
| MPAC (2, 2.0) | — | 0.979 | 4.16 | 0.933 | 4.14 | 0.897 | 4.09 | 0.864 | 4.20 |
| StealthInk (1) | — | 0.956 | 3.63 | 0.889 | 3.65 | 0.852 | 3.69 | 0.820 | 3.69 |
| StealthInk (2) | — | 0.884 | 3.56 | 0.814 | 3.65 | 0.784 | 3.74 | 0.741 | 3.74 |
| BiMark (5) | — | 0.914 | 3.64 | 0.822 | 3.64 | 0.784 | 3.62 | 0.759 | 3.63 |
| BiMark (10) | — | 0.956 | 3.63 | 0.876 | 3.59 | 0.837 | 3.66 | 0.800 | 3.58 |
| BiMark (20) | — | 0.975 | 3.53 | 0.932 | 3.51 | 0.900 | 3.61 | 0.869 | 3.59 |
| HeRo[1] (2) | — | 0.992 | 3.60 | 0.979 | 3.68 | 0.958 | 3.65 | 0.939 | 3.65 |
| HeRo[1] (3) | — | **0.994** | 3.61 | **0.989** | 3.63 | **0.972** | 3.68 | **0.954** | 3.68 |
| HeRo[2] (1,1) | ✓ | 0.964 | 3.59 | 0.912 | 3.61 | 0.875 | 3.64 | 0.842 | 3.64 |
| HeRo[2] (2,1) | ✓ | 0.976 | 3.66 | 0.941 | 3.65 | 0.900 | 3.64 | 0.870 | 3.64 |
| HeRo[2] (2,2) | ✓ | **0.985** | 3.63 | **0.945** | 3.59 | **0.916** | 3.68 | **0.876** | 3.61 |

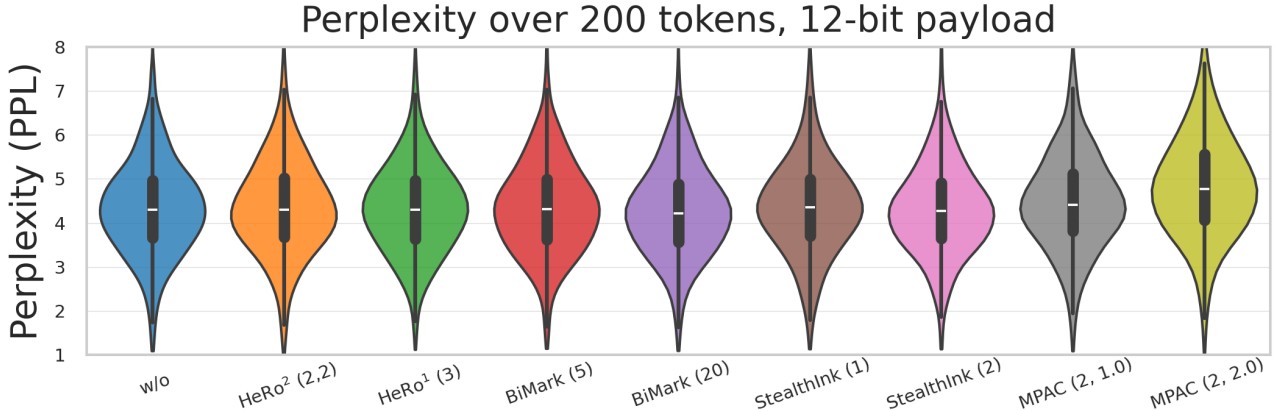

*Figure 7.* Violin plots of perplexity (PPL) for `Llama2-7B` generations on `C4` under multi-bit watermarking methods with a 12-bit payload and the unwatermarked baseline (w/o), computed over 200 generated tokens.

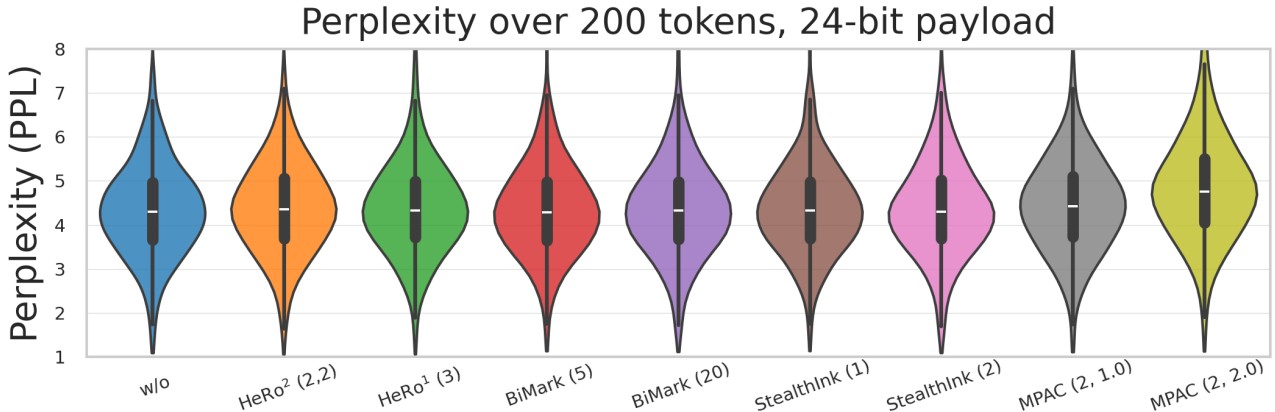

*Figure 8.* Violin plots of perplexity (PPL) for `Llama2-7B` generations on `C4` under multi-bit watermarking methods with a 24-bit payload and the unwatermarked baseline (w/o), computed over 200 generated tokens.

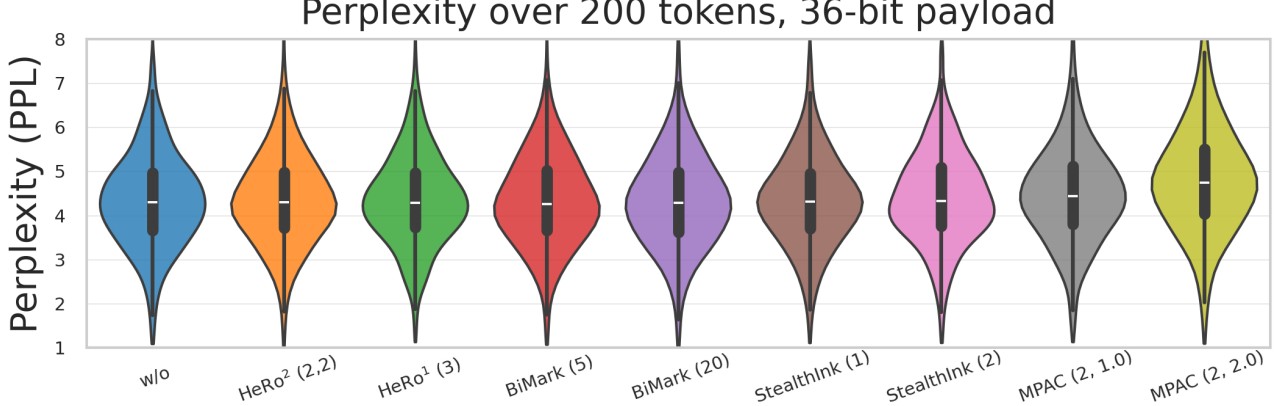

*Figure 9.* Violin plots of perplexity (PPL) for `Llama2-7B` generations on `C4` under multi-bit watermarking methods with a 36-bit payload and the unwatermarked baseline (w/o), computed over 200 generated tokens.

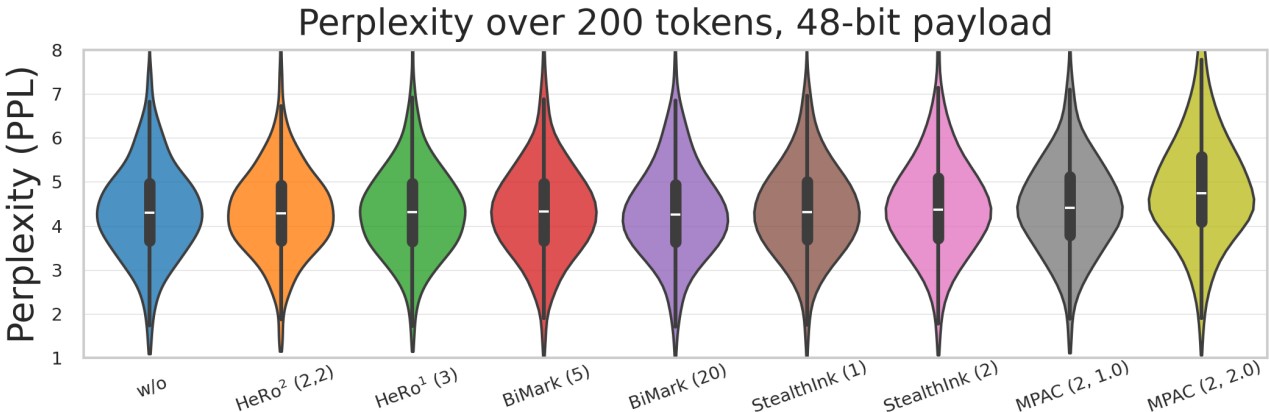

*Figure 10.* Violin plots of perplexity (PPL) for `Llama2-7B` generations on `C4` under multi-bit watermarking methods with a 48-bit payload and the unwatermarked baseline (w/o), computed over 200 generated tokens.

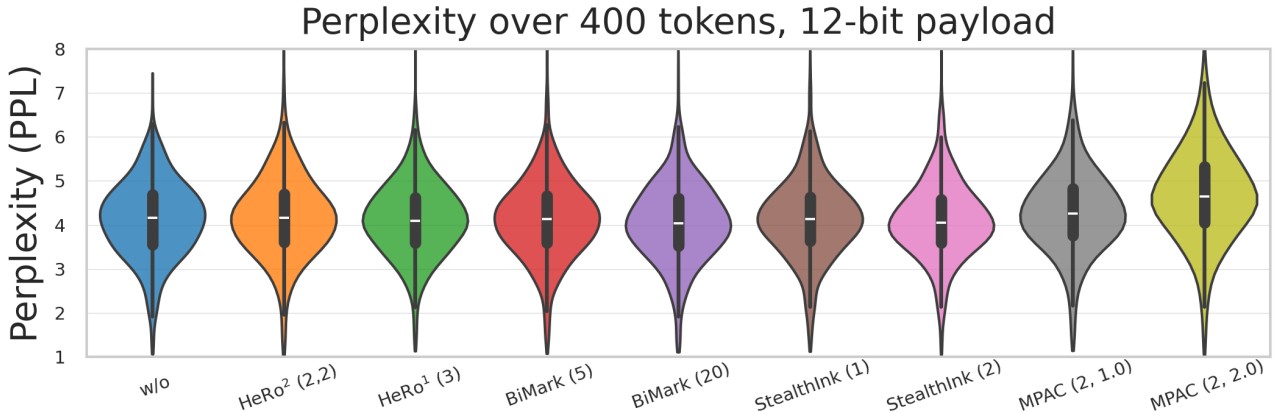

*Figure 11.* Violin plots of perplexity (PPL) for `Llama2-7B` generations on `C4` under multi-bit watermarking methods with a 12-bit payload and the unwatermarked baseline (w/o), computed over 400 generated tokens.

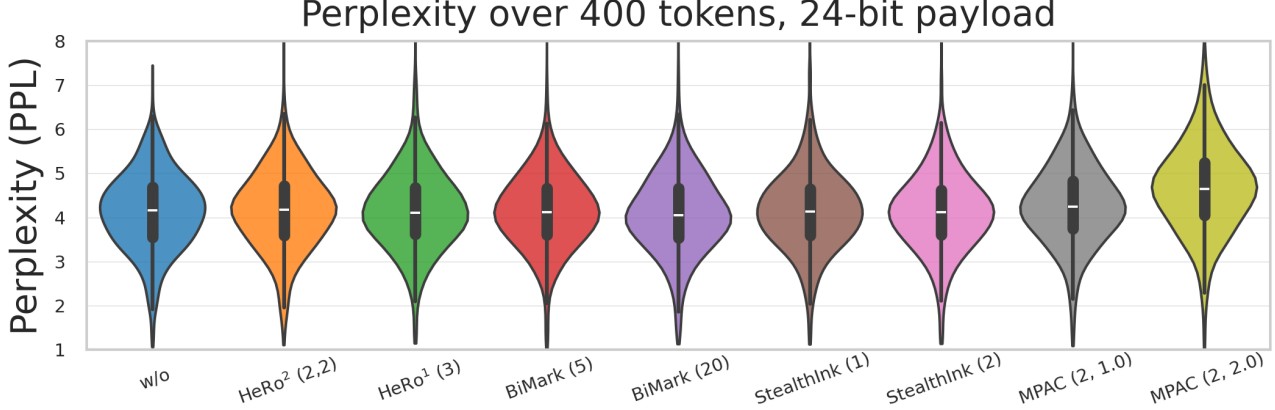

*Figure 12.* Violin plots of perplexity (PPL) for `Llama2-7B` generations on `C4` under multi-bit watermarking methods with a 24-bit payload and the unwatermarked baseline (w/o), computed over 400 generated tokens.

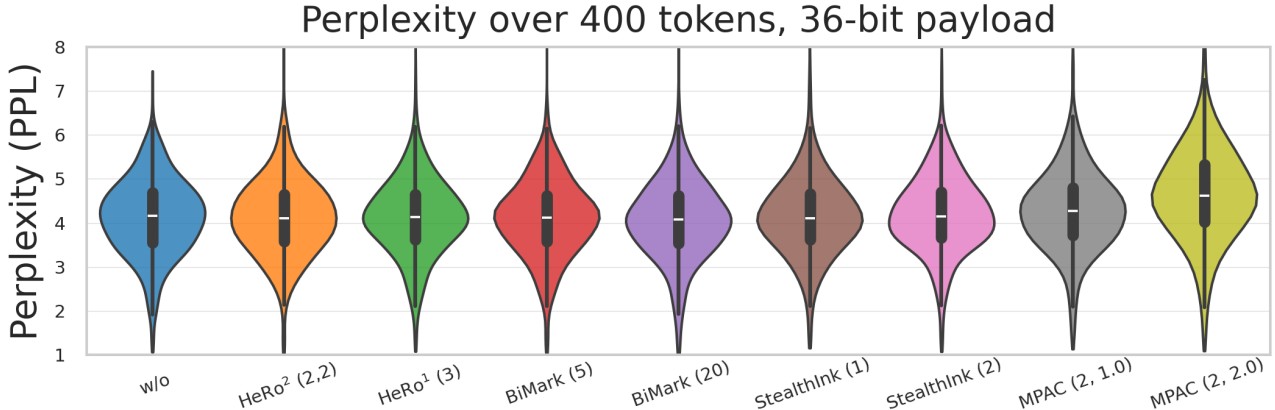

*Figure 13.* Violin plots of perplexity (PPL) for `Llama2-7B` generations on `C4` under multi-bit watermarking methods with a 36-bit payload and the unwatermarked baseline (w/o), computed over 400 generated tokens.

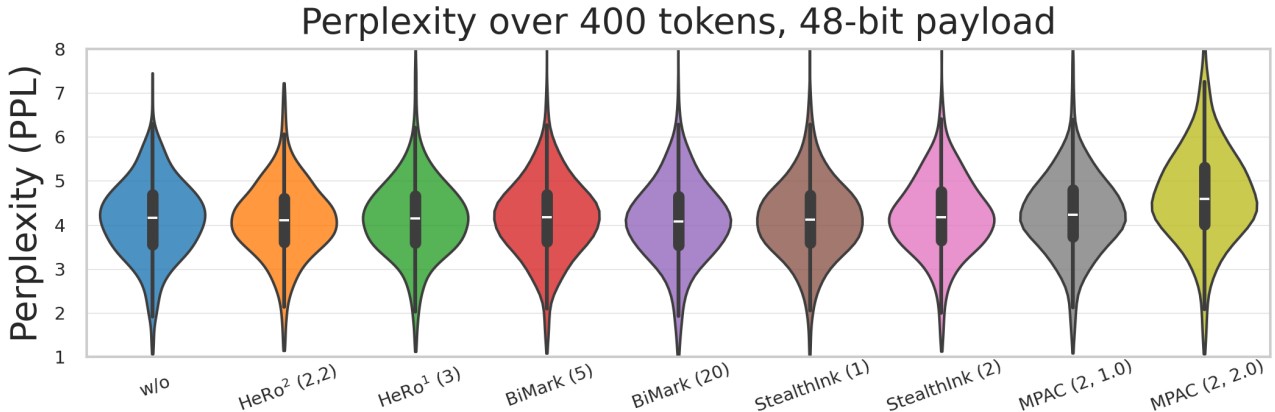

*Figure 14.* Violin plots of perplexity (PPL) for `Llama2-7B` generations on `C4` under multi-bit watermarking methods with a 48-bit payload and the unwatermarked baseline (w/o), computed over 400 generated tokens.

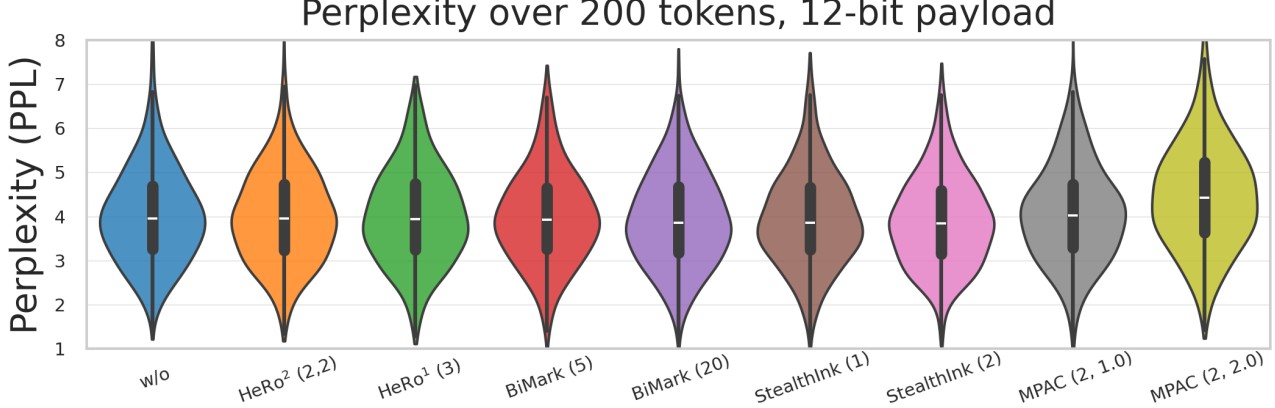

*Figure 15.* Violin plots of perplexity (PPL) for `Llama2-7B` generations on `OpenGen` under multi-bit watermarking methods with a 12-bit payload and the unwatermarked baseline (w/o), computed over 200 generated tokens.

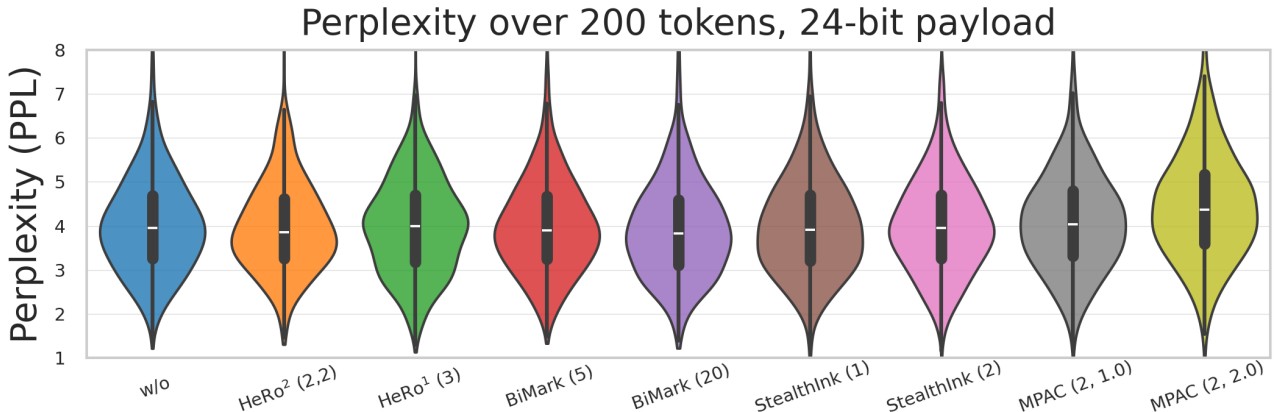

*Figure 16.* Violin plots of perplexity (PPL) for `Llama2-7B` generations on `OpenGen` under multi-bit watermarking methods with a 24-bit payload and the unwatermarked baseline (w/o), computed over 200 generated tokens.

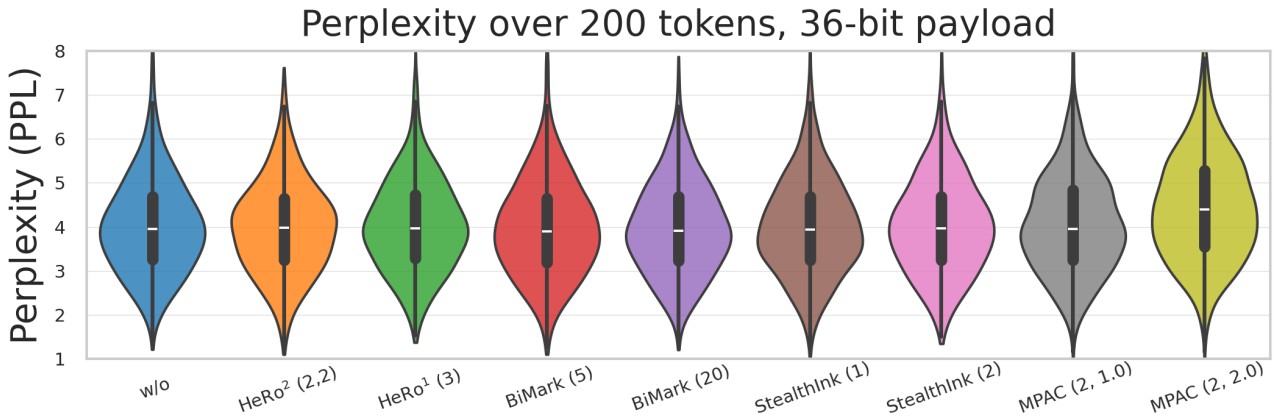

*Figure 17.* Violin plots of perplexity (PPL) for `Llama2-7B` generations on `OpenGen` under multi-bit watermarking methods with a 36-bit payload and the unwatermarked baseline (w/o), computed over 200 generated tokens.

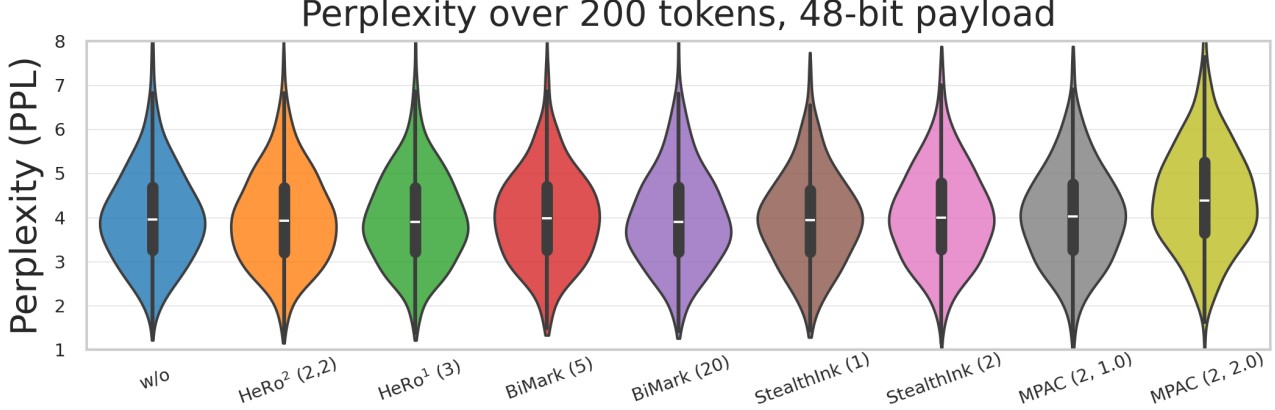

*Figure 18.* Violin plots of perplexity (PPL) for `Llama2-7B` generations on `OpenGen` under multi-bit watermarking methods with a 48-bit payload and the unwatermarked baseline (w/o), computed over 200 generated tokens.

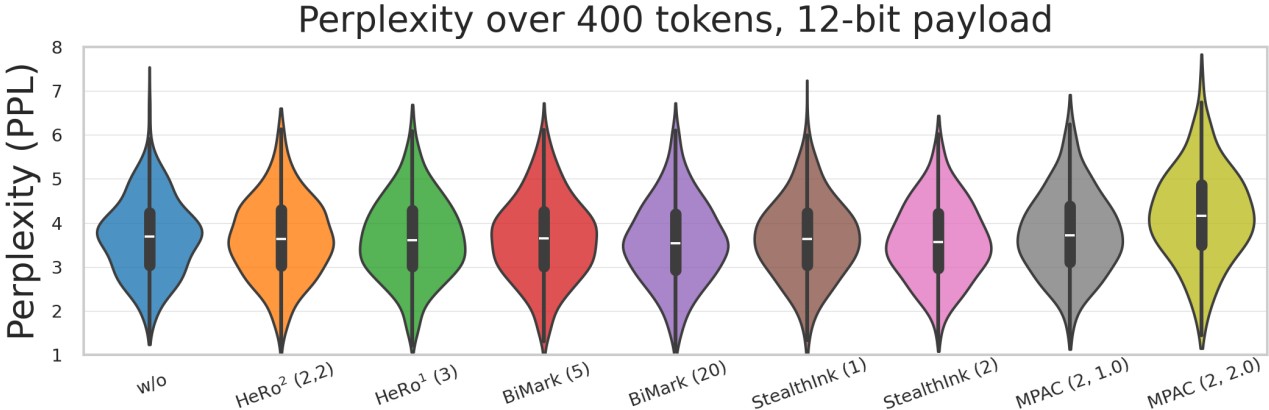

*Figure 19.* Violin plots of perplexity (PPL) for `Llama2-7B` generations on `OpenGen` under multi-bit watermarking methods with a 12-bit payload and the unwatermarked baseline (w/o), computed over 400 generated tokens.

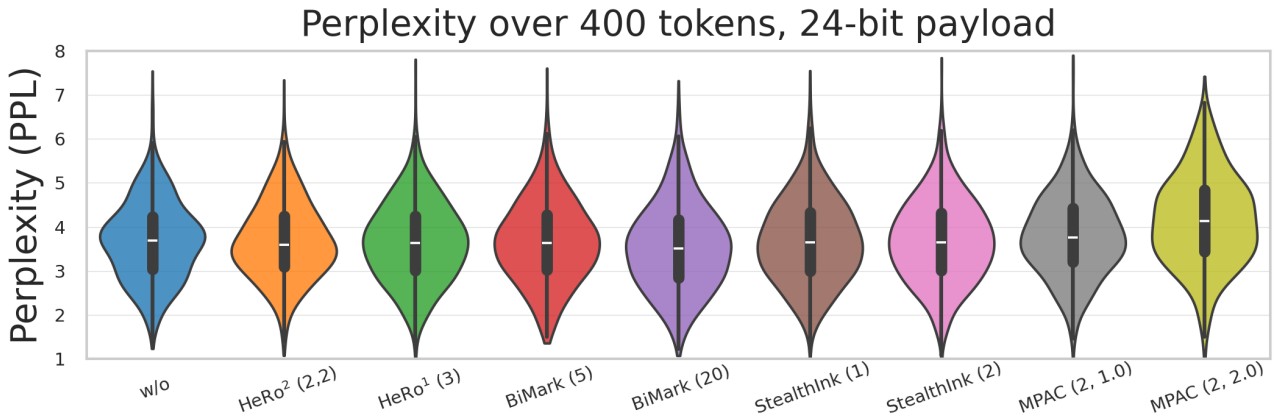

*Figure 20.* Violin plots of perplexity (PPL) for `Llama2-7B` generations on `OpenGen` under multi-bit watermarking methods with a 24-bit payload and the unwatermarked baseline (w/o), computed over 400 generated tokens.

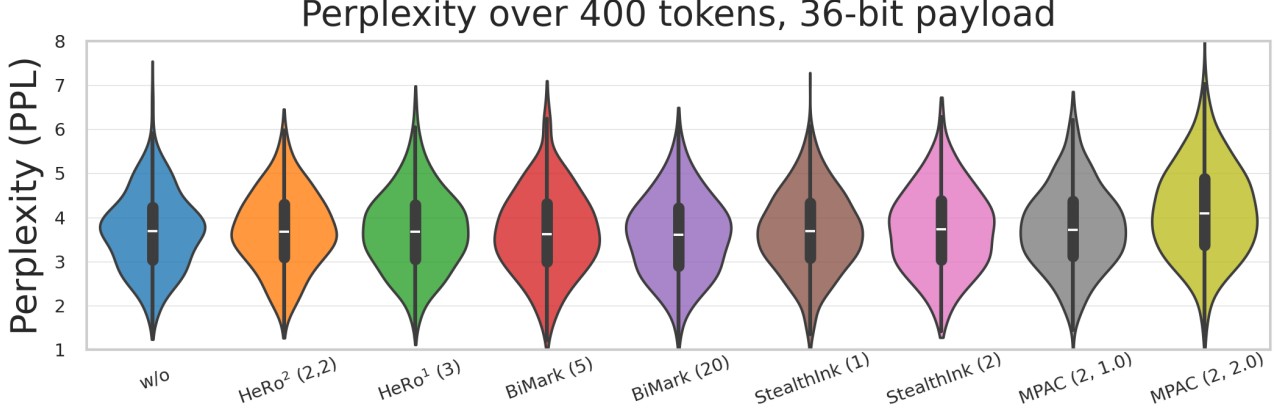

*Figure 21.* Violin plots of perplexity (PPL) for `Llama2-7B` generations on `OpenGen` under multi-bit watermarking methods with a 36-bit payload and the unwatermarked baseline (w/o), computed over 400 generated tokens.

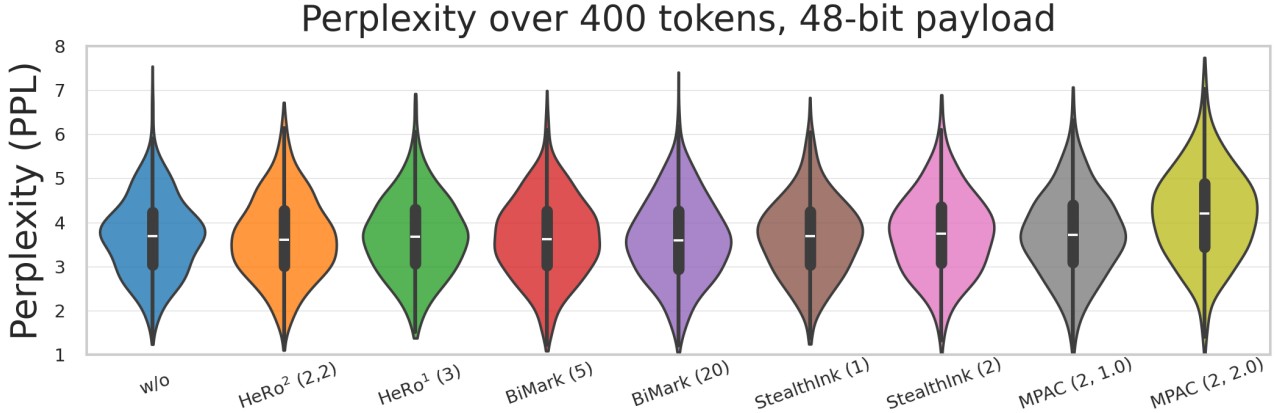

*Figure 22.* Violin plots of perplexity (PPL) for `Llama2-7B` generations on `OpenGen` under multi-bit watermarking methods with a 48-bit payload and the unwatermarked baseline (w/o), computed over 400 generated tokens.

watermarking, indicating that increasing the number of hierarchical divisions does not degrade generation quality. This is consistent with the statistical unbiasedness of our framework, which preserves the original sampling distribution.

*Table 9.* Bit accuracy (↑) under different hierarchy depths, evaluated with generation lengths of 200 and 400 tokens.

| Watermark | $n = 200$ | $n = 400$ |
|---|---|---|
| $\text{HeRo}^1(8)$ | 0.997 | 0.998 |
| $\text{HeRo}^2(4, 4)$ | 0.997 | 0.998 |
| $\text{HeRo}^4(2, 2, 2, 2)$ | 0.963 | 0.988 |
| $\text{HeRo}^8(1, 1, 1, 1, 1, 1, 1, 1)$ | 0.860 | 0.919 |

*Table 10.* Perplexity under different hierarchy depths, evaluated with generation lengths of 200 and 400 tokens.

| Watermark | $n = 200$ | $n = 400$ |
|---|---|---|
| w/o | 4.31 | 4.16 |
| $\text{HeRo}^1(8)$ | 4.26 | 4.13 |
| $\text{HeRo}^2(4, 4)$ | 4.31 | 4.15 |
| $\text{HeRo}^4(2, 2, 2, 2)$ | 4.31 | 4.15 |
| $\text{HeRo}^8(1, 1, 1, 1, 1, 1, 1, 1)$ | 4.30 | 4.13 |

# F. Watermarking Examples

| Text | Message |
|---|---|
| **Prompt:** | **Ground Truth** |
| I first met Amanda Vickery ten years ago - her book, 'The Gentleman's Daughter' had just been published, and she gave an interview to our local paper. Something about the interview made me think she would be good on the radio - her liveliness and her sense of fun came across, even in a print interview. I was right - when I called her, and we met for coffee, I realised that her warmth and her quick wit | Public message: 110 110 110 111 Private message: 100 001 111 011 |
| **Without Watermark:** | **Decoded:** |
| were the perfect match for radio. And I didn't realise, at that point, that she was a TV historian too - not just a historian, in fact, but a TV historian. Her programmes have made her a star on the BBC. I am delighted that Amanda has written a book - as a star TV historian, she has a lot of fans, but her book is a chance to get to know her as a historian. She writes beautifully - it's like she's been on our radio programmes, talking to you. It's as if she's just popped round to chat about her life and work. I like that kind of historian - someone who is engaging, passionate, and who writes well. I'm looking forward to reading this book. I can't resist giving you a quote from Amanda: 'I love reading, and I don't see myself giving up the habit for anything...' So I reckon she's the right person to write this book. She really does love her subject - and that shows. I have already read it, and enjoyed every word. If you want to be reminded of Amanda Vickery on Woman's Hour, there is a link below: The Gentleman's Daughter Amanda Vickery - A Short Introduction - part 1 Woman's Hour - Amanda Vickery Mary Beard writes about the history of feminism, and the BBC's 'The Vagina Monologues' Posted at 12:57 PM in Books, History, Interviews — Permalink — Comments (0) The 21st February marks the 100th anniversary of the first women voting in Britain. | Public message: 111 110 010 110 Private message: 000 011 101 100 |
| **HeRo$^2$(1,1):** | **Decoded:** |
| were the perfect match for radio. We decided we should make a show, and the next day we were off and recording in the park. This is the first interview that I ever did for Woman's Hour - and it was my first experience of interviewing on the radio. I had never met Amanda before, and I was nervous that she would think that I had no clue about what I was talking about. It helped enormously that she was so full of energy, and that she was so funny. I learned that, when interviewing for the radio, it is better to be yourself - and I also learned how to put questions in a way that makes the interviewee think that they are asking them! Amanda and I recorded the Woman's Hour show in London, but we did a lot of the pre-interview on the phone. Amanda is not a woman who sits still, and she had to be at the BBC while I was in my office at work, but we were able to set up the recording and make sure it worked by chatting on the phone, and then we got together for an hour at the end of the day, with a glass of wine, to finish the editing, before taking it over to the studio. Amanda was a joy to work with - she was open to the ideas we suggested, but she was also full of her own ideas, and she also had the wisdom to know when I should be in charge. When we finished, I felt that we had done a good interview, and that I could go on and do more. I am glad that I did. I have worked with many actors on the radio, but not so many who have also done interviews on TV. | Public message: 100 111 110 111 Private message: 100 101 111 011 |

*Table 11.* Watermarked text samples for a 24-bit payload. The prompt row shows the ground-truth public/private message. Decoded messages are reported for each generation, with incorrect bits highlighted in red.

---

**Algorithm 3** HeRo Generation Instantiated with Gumbel-Max Sampling

---

**Require:** Language model $\mathcal{M}$, initial prefix $x_{<1}$, generation length $T$, context window size $h$, PRF $\mathcal{A}$, key tables $\{\boldsymbol{\xi}^{(\ell)}\}_{\ell=1}^{L}$, chunking schedule $(K_1, \ldots, K_{L-1})$, full payload $m = (m_1, \ldots, m_K)$, position key $\xi_{\text{pos}}$

**Ensure:** Generated token sequence $x = (x_1, \ldots, x_T)$

1: Initialize context history $\mathcal{H} \leftarrow \emptyset$
2: **for** $t = 1$ **to** $T$ **do**
3:     Query $\mathcal{M}$ on the current prefix to obtain $P_t(\cdot) = P_{\mathcal{M}}(\cdot \mid x_{<t})$
4:     **if** $x_{(t-h):(t-1)} \in \mathcal{H}$ **then**
5:         Sample $x_t \sim P_t$
6:         Append $x_t$ to the running prefix
7:         **continue**
8:     **else**
9:         $\mathcal{H} \leftarrow \mathcal{H} \cup \{x_{(t-h):(t-1)}\}$
10:     **end if**
11:     $\zeta_t^{(\text{pos})} \leftarrow \mathcal{A}(x_{(t-h):(t-1)}, \xi_{\text{pos}})$
12:     $k_t \leftarrow \text{randint}(\zeta_t^{(\text{pos})}, K)$ {position allocation}
13:     $(m_{k_t}^{(1)}, \ldots, m_{k_t}^{(L)}) \leftarrow m_{k_t}$
14:     Set offset $o \leftarrow 0$, current chunk size $v \leftarrow V$
15:     **for** $\ell = 1$ **to** $L - 1$ **do**
16:         Partition $[o, o+v)$ into $K_\ell$ contiguous chunks $\mathcal{C}_1^{(\ell)}, \ldots, \mathcal{C}_{K_\ell}^{(\ell)}$
17:         $w_i^{(\ell)} \leftarrow \sum_{x \in \mathcal{C}_i^{(\ell)}} P_t(x)$ for $i = 1, \ldots, K_\ell$
18:         $\xi \leftarrow \boldsymbol{\xi}^{(\ell)}(m_{k_t}^{(\ell)})$
19:         $\zeta_t^{(\ell)} \leftarrow \mathcal{A}(x_{(t-h):(t-1)}, \xi)$
20:         Use $\zeta_t^{(\ell)}$ as the PRNG seed to generate $(U_{t,i}^{(\ell)})_{i=1}^{K_\ell}$ with $U_{t,i}^{(\ell)} \overset{\text{i.i.d.}}{\sim} \text{Uniform}(0,1)$
21:         $s_{t,\ell} \leftarrow \arg\max_i \ \log w_i^{(\ell)} - \log(-\log U_{t,i}^{(\ell)})$
22:         Update $o, v$ to the range of $\mathcal{C}_{s_{t,\ell}}^{(\ell)}$
23:     **end for**
24:     $\xi \leftarrow \boldsymbol{\xi}^{(L)}(m_{k_t}^{(L)})$
25:     $\zeta_t^{(L)} \leftarrow \mathcal{A}(x_{(t-h):(t-1)}, \xi)$
26:     Use $\zeta_t^{(L)}$ as the PRNG seed to generate $(U_{t,j}^{(L)})_{j=0}^{v-1}$ with $U_{t,j}^{(L)} \overset{\text{i.i.d.}}{\sim} \text{Uniform}(0,1)$
27:     $s_{t,L} \leftarrow \arg\max_{j=0,\ldots,v-1} \ \log P_t(o+j) - \log(-\log U_{t,j}^{(L)})$
28:     $x_t \leftarrow o + s_{t,L}$
29:     Append $x_t$ to the running prefix
30: **end for**
31: **return** $x = (x_1, \ldots, x_T)$

---

---

**Algorithm 4** HeRo Decoding Instantiated with Gumbel-Max Sampling

---

**Require:** Batch of generated sequences $x^{(1)}, \ldots, x^{(B)}$, vocabulary size $V$, context window size $h$, PRF $\mathcal{A}$, key tables $\{\boldsymbol{\xi}^{(\ell)}\}_{\ell=1}^{L}$, chunking schedule $(K_1, \ldots, K_{L-1})$, bits per level $(b_1, \ldots, b_\ell)$, position key $\xi_{\text{pos}}$, number of segments $K$

**Ensure:** Decoded payloads $\hat{m}^{(1)}, \ldots, \hat{m}^{(B)}$

1: Convert the batch into decoding pairs $\{(x_{(t-h):(t-1)}, x_t)\}_{t=1}^{T}$, where each pair contains one context window and the corresponding next token
2: {Stage 1: compute token-level evidence}
3: Reconstruct position allocation $\{k_t\}_{t=1}^{T}$ in parallel from $\{x_{(t-h):(t-1)}\}_{t=1}^{T}$ using $\xi_{\text{pos}}$
4: **for** $\ell = 1$ **to** $L - 1$ **do**
5:     Compute $\{s_{t,\ell}\}_{t=1}^{T}$ in parallel by locating each token $x_t$ in the level-$\ell$ chunk partition
6: **end for**
7: Compute $\{s_{t,L}\}_{t=1}^{T}$ in parallel from the final selected chunk
8: **for** $\ell = 1$ **to** $L$ **do**
9:     **for** $a = 0$ **to** $2^{b_\ell} - 1$ **do**
10:         $\xi \leftarrow \boldsymbol{\xi}^{(\ell)}(a)$
11:         Compute $\{U_t^{(\ell)}(a)\}_{t=1}^{T}$ in parallel, where

$$U_t^{(\ell)}(a) \leftarrow \left[\mathcal{A}(x_{(t-h):(t-1)}, \xi)\right]_{s_{t,\ell}}$$

12:         Compute $\{E_t^{(\ell)}(a)\}_{t=1}^{T}$ in parallel, where

$$E_t^{(\ell)}(a) \leftarrow -\log(1 - U_t^{(\ell)}(a))$$

13:     **end for**
14: **end for**
15: {Stage 2: aggregate evidence and decode the message}
16: **for** $\ell = 1$ **to** $L$ **do**
17:     **for** $a = 0$ **to** $2^{b_\ell} - 1$ **do**
18:         Compute $\text{Score}_{i,k}^{(\ell)}(a)$ by summing $E_t^{(\ell)}(a)$ over all pairs from sequence $x^{(i)}$ with $k_t = k$, for all $i = 1, \ldots, B$ and $k = 1, \ldots, K$ in parallel
19:     **end for**
20:     Compute $\hat{m}_{i,k}^{(\ell)} \leftarrow \arg\max_a \text{Score}_{i,k}^{(\ell)}(a)$ for all $i = 1, \ldots, B$ and $k = 1, \ldots, K$ in parallel
21: **end for**
22: Set $\hat{m}^{(i)} \leftarrow \left((\hat{m}_{i,1}^{(1)}, \ldots, \hat{m}_{i,1}^{(L)}), \ldots, (\hat{m}_{i,K}^{(1)}, \ldots, \hat{m}_{i,K}^{(L)})\right)$ for each $i = 1, \ldots, B$
23: **return** $\hat{m}^{(1)}, \ldots, \hat{m}^{(B)}$

---

