# OpenReview forum: "Selective Disclosure Watermarking for Large Language Models"
_ICML.cc/2026/Conference — ICML 2026 regular_

### Official Review · Reviewer_CcxH · 2026-02-24

**Soundness:** 3
**Presentation:** 3
**Significance:** 3
**Originality:** 3
**Overall Recommendation:** 5
**Confidence:** 3

**Summary:**

This paper proposes "Hierarchical Vocabulary Routing" (HeRo), a multi-bit watermarking framework for LLM-generated text that uniquely supports "selective disclosure", allowing different verifiers to decode different levels of embedded metadata from the same generated text. The method recursively partitions the vocabulary and performs hierarchical chunk routing tied to keys/messages. HeRo is statistically unbiased, meaning that the token marginals match the base model distribution. Experimental results indicate high bit decoding accuracy, minimal impact on perplexity, low computational overhead, and fast decoding. Further, HeRo is more robust to text modifications than the other evaluated schemes.

**Compliance With Llm Reviewing Policy:**

Affirmed.

**Final Justification:**

The authors have addressed my concerns, particularly regarding the limited robustness evaluation. Since I haven't identified any other weaknesses, I'm satisfied to raise my score.

**Key Questions For Authors:**

1. How robust is the scheme to stronger and more realistic edits (e.g., paraphrasing via DIPPER)?
2. Does degradation differ by hierarchy level?
3. For latency comparisons, how much of the advantage is algorithmic (and not implementation-specific)?

**Limitations:**

The impact statement discusses risks and corresponding methods of mitigation. However, limitations around robustness to realistic edits should be stated more concretely.

**Strengths And Weaknesses:**

Strengths:
Selective disclosure is a practically relevant capability largely missing from prior watermarking schemes. The hierarchical routing abstraction is simple yet powerful. Experimental results are strong, regarding not only detectability, but also quality preservation and robustness to edits. The scheme is also very efficient in both generation overhead and decoding complexity.

Weaknesses:
The robustness evaluation is limited to random token replacement and round-trip translation. Stronger edits (LLM paraphrasing, insertion/deletion of text spans, summarization, one-way translation) are not evaluated.

Formatting and Presentation:
The paper is formatted well, and the results are presented clearly. However, it would benefit from replacing raster graphics with vector graphics.

---

> ### Author Rebuttal · Authors · 2026-03-31
>
> We thank the reviewer for the supportive assessment and acknowledge our contribution ("**Selective disclosure is a practically relevant capability largely missing from prior watermarking schemes**."). We hope our responses address the concerns.
> ## Response to Q1: Robustness under stronger and realistic edits
> Following the reviewer's suggestion, we conduct additional experiments comparing the robustness of methods under stronger and more realistic edits. The first attack is LLM paraphrasing via DIPPER, where we vary (lex_diversity, order_diversity) across $(20, 0), (0, 20), (20, 20)$. These two parameters control the amount of lexical diversity and reordering in the paraphrases, respectively. The second attack consists of insertion and deletion of text spans, where we vary the perturbation rate at $0.05$, $0.10$ and $0.20$. We report bit accuracy under both attacks on the C4 dataset using Llama2-7B with a 24-bit payload, across text lengths of $200$ and $400$ tokens. The results indicate that the bit accuracy across all methods degrades as the intensity of paraphrasing increases. Our method consistently achieves the highest decoding accuracy among all evaluated methods.
>
> |Bit Accuracy (n=200)|DIPPER|DIPPER|DIPPER|Insertion|Insertion|Insertion|Deletion|Deletion|Deletion|
> |---|---|---|---|---|---|---|---|---|---|
> ||(20,0)|(0,20)|(20,20)|0.05|0.1|0.2|0.05|0.1|0.2|
> |MPAC (2, 2.0)|0.669|0.794|0.646|0.814|0.754|0.648|0.827|0.766|0.663|
> |StealthInk (2)|0.582|0.665|0.573|0.684|0.630|0.559|0.690|0.643|0.575|
> |BiMark (20)|0.703|0.817|0.691|0.838|0.778|0.677|0.845|0.793|0.692|
> |HeRo $^1$(3)|**0.800**|**0.931**|**0.778**|**0.952**|**0.906**|**0.782**|**0.958**|**0.920**|**0.798**|
> |HeRo $^2$(2,2)|**0.723**|**0.848**|**0.701**|**0.868**|**0.809**|**0.707**|**0.879**|**0.825**|**0.710**|
>
> |Bit Accuracy (n=400)|DIPPER|DIPPER|DIPPER|Insertion|Insertion|Insertion|Deletion|Deletion|Deletion|
> |---|---|---|---|---|---|---|---|---|---|
> ||(20,0)|(0,20)|(20,20)|0.05|0.1|0.2|0.05|0.1|0.2|
> |MPAC (2, 2.0)|0.758|0.895|0.714|0.911|0.853|0.726|0.920|0.854|0.711|
> |StealthInk (2)|0.651|0.767|0.642|0.773|0.709|0.618|0.776|0.713|0.607|
> |BiMark (20)|0.787|0.902|0.777|0.915|0.862|0.751|0.921|0.870|0.748|
> |HeRo $^1$(3)|**0.898**|**0.981**|**0.853**|**0.988**|**0.970**|**0.876**|**0.989**|**0.971**|**0.866**|
> |HeRo $^2$(2,2)|**0.805**|**0.926**|**0.766**|**0.938**|**0.890**|**0.781**|**0.941**|**0.897**|**0.768**|
>
> ## Response to Q2: Performance variation across hierarchy levels
> Please refer to our response to Q7 from Reviewer `Ytbb`, where we provide detailed results across different hierarchy depths. In summary, deeper layers exhibit lower detection accuracy due to reduced per-layer signal, but the gap narrows with longer token sequences.
> ## Response to Q3: Algorithmic vs. implementation contributions to latency
> A substantial part of the latency advantage is algorithmic rather than implementation-specific.
>
> Our contribution is to identify a structural asymmetry between watermark generation and detection. In the generation phase, the algorithm constructs the full pseudorandom variable $\zeta$ to inject the watermark signal. Prior approaches have typically treated detection as if it likewise required reconstructing the full pseudorandom variable. Our key observation is that this is unnecessary. For the detection algorithm, it suffices to recover only the random variate $\zeta_s$ associated with the observed token index $s$, and the information corresponding to $s'\neq s$ is redundant. This significantly reduces the per-step decoding complexity to $O(\sum_{\ell=1}^L 2^{b_\ell})$, which is independent of the vocabulary size.
>
> We will release the code for better contribution.
> ## About Formatting and Presentation
> Thank you for the suggestion. We will replace raster graphics with vector graphics in the revised manuscript for better presentation.

---

> > ### Author Rebuttal · Reviewer_CcxH · 2026-04-03
> >
> > Thank you for addressing my concerns and for your detailed response, particularly regarding the limited robustness evaluation. Since I haven't identified any other weaknesses, I'm satisfied to raise my score.

---

> > > ### Author Response · Authors · 2026-04-04
> > >
> > > We sincerely thank the reviewer for the helpful suggestions and the positive feedback. We are glad that our responses have fully addressed your concerns.

---

### Official Review · Reviewer_Ytbb · 2026-03-03

**Soundness:** 4
**Presentation:** 3
**Significance:** 4
**Originality:** 4
**Overall Recommendation:** 4
**Confidence:** 4

**Summary:**

The authors address the multi-bit watermarking problem in large language models (LLMs) by proposing a novel method for granular access control. Specifically, they introduce Hierarchical Vocabulary Routing (HeRo), a framework that leverages hierarchical data embedding. HeRo recursively partitions the LLM vocabulary into a nested tree structure, embedding distinct payload segments across various hierarchical layers. Consequently, verifiers can decode only the specific payload layers for which they possess the corresponding watermark keys. The authors provide theoretical guarantees of statistical unbiasedness and selective disclosure, supported by empirical evaluations that demonstrate high detection accuracy, preserved text quality, and computational efficiency relative to baseline methods.

**Compliance With Llm Reviewing Policy:**

Affirmed.

**Final Justification:**

The author has addressed my concerns, and my decision on the paper is weak accept.

**Key Questions For Authors:**

Questions for the Authors:

1. In the zero-bit case, the standard EMS method first generates watermark keys (typically longer than length of sequence to be detected) to generate the watermarked text, and the evidence function in summation form subsequently takes the maximum over all watermark keys subsequence. Did the authors implement this exact procedure for each layer/level/stage? Furthermore, is the length of the watermark keys consistent across all layers?

2. Does the proposed method generate every token $x_t$ using the entire message sequence $m = (m_1, \dots, m_K)$? Please confirm if this understanding is correct.

3. Regarding the proof of Theorem 4.1, please provide further explanation for the equations at the bottom of page 12 (following "For the second term, we have..."). This appears to be a core step in the proof, but the mathematical derivation is not immediately obvious.

4. Is the nested chunk path, i.e. the partition of the vocabulary across all levels, fixed or dynamically determined?

5. In the simplest setting (the zero-bit case with only one level, where the goal is solely to detect the exist of the watermark), does the proposed method exactly reduce to the standard EMS framework?

6. I require clarification regarding the "total payload size." For example, in Section 5.1, it is unclear why the total payload size is 24 for different values of $b_1$ and $b_2$. Based on my understanding, if $b_1 = 1$ and $b_2 = 1$, the total payload size should be $2 \times 2 = 4$. Could you please clarify this calculation?

7. Although the discussion says that "characterizing this trade-off and identifying fundamental limits between selective disclosure and detectability remain open problems," it would be highly beneficial to empirically demonstrate the method's performance across deeper layers. For instance, fixing $K = 8$ with $m = (m_1, \dots, m_K)$ and $m_i \in \{0, 1\}$, it would be insightful to compare results across 1 layer (containing all messages), 2 layers (4 messages per layer), 4 layers (2 messages per layer), and 8 layers (1 message per layer).

**Limitations:**

Yes.

**Strengths And Weaknesses:**

The introduction of "selective disclosure" to LLM watermarking is highly relevant for real-world auditing applications, representing a novel and practical contribution. Whereas most current multi-bit methods rely on red-green list-based approaches, the authors propose an EMS-based method that guarantees the statistical unbiasedness of the watermark insertion. Furthermore, the proposed methodology is supported by rigorous theoretical guarantees.

However, the mathematical notation is overly complex, making the paper difficult to follow. While I understand the authors' intent to formulate their method with broad generality rather than restricting it to the EMS framework, this level of abstraction significantly hinders readability. Although the high-level concept is intuitive and conceptually sound, the precise implementation details remain obscure. To improve comprehension and reproducibility, I strongly recommend adding an appendix section that provides a detailed, step-by-step guide on how to implement the proposed method within the EMS framework for both encoding and decoding parts.

---

> ### Author Rebuttal · Authors · 2026-03-31
>
> We sincerely thank the reviewer for the positive and thoughtful feedback. We address each question below.
> ## Response to Q1: Key generation and dimensionality across layers
> 1. We do not precompute a long global watermark-key sequence. Instead, we use a context-dependent key derivation strategy: at each generation step, the pseudorandom variable at level $\ell$ is derived on-the-fly by applying a PRF to the local context window together with level-$\ell$ secret message and a secret key querying table.
>
> 2. The length of the pseudorandom variable is not uniform across levels. Under Gumbel-Max sampling, the pseudorandom variable at routing stage $\ell$ has dimension $K_\ell$ determined by the chunking schedule. Because the chunking schedule $(K_1,\ldots,K_{L-1})$ may assign different numbers of chunks to different levels, the pseudorandom variable lengths generally differ across levels. At the final stage $L$, the pseudorandom variable has dimension equal to the size of the remaining candidate set $|C^{(L-1)}|\approx|V|/\prod_{\ell=1}^{L-1}K_\ell$​.
> ## Response to Q2: Whether each token uses the full message
> No. Each token is not generated using the entire message sequence. We follow the position allocation strategy used in multi-bit watermarking [1,2,3]. The full payload $m$ is first partitioned into $K$ segments $m=(m_1,\cdots,m_K)$, and each token position $t$ is assigned to a single segment via a rule $p(t)\in\lbrace 1,\dots,K\rbrace$. Therefore, each token $x_t$ encodes one segment $m_{p(t)}$, rather than the entire message.
>
> In HeRo, this assigned segment is further decomposed across hierarchical levels as $(m_{p(t)}^{(1)},\cdots,m_{p(t)}^{(L)})$. The generation of $x_t$ then proceeds layer by layer following Algorithm 1: at each level $\ell$, the routing decision depends only on the corresponding sub-message $m_{p(t)}^{(\ell)}$.From the perspective of the hierarchical routing process, each token induces a path $\mathcal{V}=C^{(0)}\supset C^{(1)}\supset\cdots\supset C^{(L)}=\lbrace x_t\rbrace$, where the routing decision at level $\ell$ is controlled only by $m_{p(t)}^{(\ell)}$.
>
> Therefore, each token encodes only a portion of the payload, distributed across levels.
>
> [1] Advancing Beyond Identification: Multi-bit Watermark for Large Language Models, ACL 2024.
>
> [2] StealthInk: A Multi-bit and Stealthy Watermark for Large Language Models, ICML 2025.
>
> [3] BiMark: Unbiased Multilayer Watermarking for Large Language Models, ICML 2025.
> ## Response to Q3: Clarification of the proof (Theorem 4.1)
> Please refer to our response to Q3 from `BExV`, where we provide a detailed clarification of the key step in the proof.
> ## Response to Q4: Fixed partition vs. dynamic chunk path
> The **partition rule is fixed**, while the **nested chunk path is dynamic**.
>
> The **partition rule** is fixed: at each stage, the candidate vocabulary set is partitioned into contiguous chunks by a deterministic rule (see Page 4 footnote). This is why the decoder can deterministically recover the nested chunk path after observing the token. The **nested chunk path** is dynamic since the chunk selected at each level depends on sampling.
> ## Response to Q5: Relation to EMS in the single-layer case
> Yes. When $L=1$, there is no routing stage, and the framework reduces to the underlying unbiased sampler. In particular, if instantiated with EMS, the zero-bit case recovers the standard EMS framework.
> ## Response to Q6: Interpretation of total payload size
> By "total payload size," we mean the length of the full metadata embedded in one generation, not the number of bits contributed by a single token. For example, $b_1=b_2=1$ means we assign each token $b_1+b_2=2$ bits message, the full 24-bit payload is segmented across $K=24/2=12$ positions and embedded using position allocation accordingly.
> ## Response to Q7: Performance under deeper hierarchies
> Following the reviewer’s recommendation, we conduct additional experiments under 8-bit payload per token, comparing four hierarchical settings: HeRo $^1$(8), HeRo $^2$(4,4), HeRo $^4$(2,2,2,2), and HeRo $^8$(1,1,1,1,1,1,1,1), where Hero $^L(b_1,\cdots,b_L)$ denotes an $L$-level hierarchy in which the $l$-th layer discloses $b_l$-bits.
> |Bit Accuracy|n=200|n=400|
> |-|-|-|
> |HeRo $^1$(8)|0.997|0.998|
> |HeRo $^2$(4,4)|0.997|0.998|
> |HeRo $^4$(2,2,2,2)|0.963|0.988|
> |HeRo $^8$(1,1,1,1,1,1,1,1)|0.860|0.919|
>
> The results suggest that detection generally becomes more challenging as the hierarchy becomes deeper. We observe that the performance gap across settings becomes smaller as the token budget increases. More broadly, the performance depends jointly on the hierarchy depth and the partition structure across levels, and there remains room to further optimize deeper configurations.
> ## About the readability and reproducibility
> We appreciate the suggestion. We will improve the presentation and include detailed implementation steps in the appendix to enhance clarity. We also plan to release code to facilitate reproducibility.

---

> > ### Author Rebuttal · Reviewer_Ytbb · 2026-04-03
> >
> > Thank you for your explanation. I am still unclear about how the proposed method reduces to the EMS framework. In particular, the authors state that the method reduces to EMS in the zero-bit setting. In this case, is $b_1 = 1$ or $b_1 = 0$? Moreover, how is detection performed under this setting?
> >
> > I believe this question is closely related to watermark detection in the multi-bit setting. Suppose the goal is simply to determine whether a given text is watermarked. In this framework, should we set $m \in \{0, 1\}$, where $m = 0$ indicates no watermark and $m = 1$ indicates the presence of a watermark, and then generate the watermark key via $\xi(m)$? In the detection stage, do we then test whether $m = 0$ or $m = 1$? If this understanding is correct, it seems that in both cases ($m = 0$ and $m = 1$), the generated text is still produced using the watermarking mechanism, which raises a conceptual concern. Could the authors please clarify this point?

---

> > > ### Author Response · Authors · 2026-04-04
> > >
> > > Thank you for the quick follow-up.
> > >
> > > ### Standard EMS as a Zero-Payload Special Case
> > > The standard EMS reduction corresponds to the **single-level**, **zero-payload** case. If the total payload length is $B$-bit, then the message space is $\\mathcal{M}_B=\\{0,1\\}^B$. In the zero-bit case ($B=0$), this reduces to
> > > $$\\mathcal{M}_0=\\{0,1\\}^0=\\{\\varepsilon\\},$$
> > >
> > > a singleton containing only the empty bitstring. Therefore, there is no nontrivial message variable to encode, and the message-bearing notation in our multi-bit formulation (e.g., $b_{\\ell}$) degenerates in this special case rather than defining a 1-bit problem. Correspondingly, the secret table contains only a **single** key, associated with the unique element of $\\mathcal{M}_0$. The generation procedure reduces to the underlying unbiased sampler with this key. When that sampler is instantiated as EMS, this recovers exactly the standard zero-bit EMS framework.
> > >
> > > ### Binary Watermark Detection
> > > We clarify that our framework supports binary watermark detection when the goal is only to determine whether a text is watermarked. The results are reported in Table 5, where we measure true positive rate (TPR) at fixed target false positive rate (FPR).
> > >
> > > In the general $L$-level case, the decoder first computes candidate-evidence scores described in Section 4.2 and aggregates them, yielding multi-level statistics of the form $\\{\\mathrm{Agg}(\\{E_t^{(\\ell)}(a)\\})\\}_{l,a}.$
> > >
> > > These aggregated evidence scores are then mapped to a scalar detection statistic $Y$, and the text is identified as watermarked if $Y\\ge\\gamma_\\alpha$, where $\\gamma_\\alpha$ is a threshold calibrated to achieve a target FPR $\\alpha$. As an illustration, consider the single-level ($L=1$), $B$-bit setting with EMS as the sampling rule. For each candidate message $a$, the detector computes a per-token evidence score using the Aaronson score $E_t^{(1)}(a) = -\\log\\big(1-\\zeta_t^{(a)}(s_t)\\big),$ aggregated as $\\mathrm{Agg}\\big(\\{E_t^{(1)}(a)\\}\\big) = \\sum_{t=1}^T E_t^{(1)}(a)$. Under the null hypothesis that the text is not watermarked, we have $X_a=\\mathrm{Agg}\\big(\\{E_t^{(1)}(a)\\}\\big)\\sim\\Gamma(T)$, where $T$ is the number of tokens. We then define the statistic $Y=\max_{a\\in\\mathcal{M}_B}X_a,$
> > >
> > > and let $\\gamma_\\alpha$ be the $(1-\\alpha)$-quantile of the null distribution of $Y$. The decision rule "identify the text watermarked if $Y\\ge\\gamma_\\alpha$" defines a level-$\\alpha$ test.
> > >
> > > We hope these clarifications help resolve the concerns.

---

### Official Review · Reviewer_sHdd · 2026-03-13

**Soundness:** 1
**Presentation:** 3
**Significance:** 2
**Originality:** 1
**Overall Recommendation:** 3
**Confidence:** 4

**Summary:**

This paper studies multi-bit watermarking for LLM-generated text and highlights the limitation of existing methods in supporting selective disclosure of embedded metadata. To address this issue, the authors propose HeRo (Hierarchical Vocabulary Routing), a framework that hierarchically partitions the vocabulary and distributes watermark information across multiple layers, allowing different verifiers to decode only the metadata corresponding to their access level. The method preserves the unbiasedness of the sampling process, thereby maintaining generation quality. Experimental results show that HeRo enables fine-grained access control while achieving reliable detection with low latency.

**Compliance With Llm Reviewing Policy:**

Affirmed.

**Final Justification:**

most concerns resolved, the novelty concern still remain.

> (a) We respectfully point out that our method and prior multilayer reweighting approaches rely on two structurally distinct mechanisms, and therefore should be considered separately.

> Single sampling. In methods [1] and [2], multiple layers of probability reweighting are applied over the full vocabulary, followed by a single sampling event to generate the token.

> Multiple sampling. In HeRo, each token is generated each token through $L$ sequential sampling stages. At routing stage $\ell$, the sampler draws from a $K_\ell$​-way categorical distribution over aggregated chunk probabilities (meta-tokens), and the selected chunk defines the candidate set for stage $\ell+1$. This induces a nested structure $ \mathcal{V} = C^{(0)} \supset C^{(1)} \supset \cdots \supset C^{(L)}=\{x_t\}, $ which does not exist in prior methods.

I respectfully disagree with this point. In [1], the intuition of tournament sampling is indeed multiple sampling, the authors of [1] then simplify the multiple sampling to single sampling for compute efficiency consideration.

**Key Questions For Authors:**

See weaknesses

**Limitations:**

Yes

**Strengths And Weaknesses:**

## Strengths:

The proposed multi-layer watermarking framework preserves the unbiasedness of the underlying sampling process, which is an important property for maintaining the original generation distribution and avoiding degradation in text quality.

The paper provides comprehensive experimental evaluations covering watermark detectability, robustness, and generation quality, which together offer a thorough empirical assessment of the proposed method and demonstrate its effectiveness across multiple metrics.

## Weaknesses:

The main idea of multi-layer watermarking is not novel. Prior work has already explored similar designs: [1] introduces a multi-layer tournament sampling scheme, and [2] further generalizes this idea into a broader multi-layer watermarking framework. The structure and design of the proposed method seem largely consistent with the framework described in [2]. As a result, the novelty of the proposed approach is limited, and the paper would benefit from a clearer discussion and comparison with these closely related works.

[1] Scalable watermarking for identifying large language model outputs, Nature 2024.

[2] An Ensemble Framework for Unbiased Language Model Watermarking, ICLR 2026.

---

> ### Author Rebuttal · Authors · 2026-03-31
>
> We thank the reviewer for the feedback. However, we believe the main concern regarding lack of novelty is based on a mischaracterization of our contribution. We would like to clarify that our contribution is not another instance of "multi-layer watermarking." While the cited works [1, 2] and HeRo both involve layered mechanisms at a high level, they address fundamentally different problems. In our view, the label "multi-layer watermarking" is too broad in this context and obscures the key distinction of our work: the novel selective disclosure capability enabled by our hierarchical design.
>
> [1] Scalable watermarking for identifying large language model outputs, Nature 2024.
>
> [2] An Ensemble Framework for Unbiased Language Model Watermarking, ICLR 2026.
>
> [3] BiMark: Unbiased Multilayer Watermarking for Large Language Models, ICML 2025.
>
> ## Prior work [1,2]: binary detection; HeRo: selective disclosure
> The works [1,2] cited in the review focus on **binary watermark detection**, where the goal is to determine whether a text is watermarked. Their "multi-layer" mechanisms apply multiple reweightings over the vocabulary to strengthen a single binary detection signal. One of our baselines, [3], utilizes multi-layer reweighting idea to strengthen **multi-bit watermark watermarking**. However, it still follows an **all-or-nothing disclosure model**: verifying any part of the watermark necessarily reveals the entire embedded message. In fact, all of the above methods face this same limitation.
>
> In contrast, HeRo fundamentally differs by enabling **partial, role-based decoding**, which requires a different design rather than repeated reweighting. Specifically, HeRo uses hierarchical vocabulary routing to progressively restrict the candidate token set, so that different layers encode different portions of the payload. As a result, verifiers with different authorization levels can recover different subsets of the embedded metadata.
> ## Our contributions
> Our contributions go beyond a new watermarking variant and span three dimensions:
>
> 1. **Problem formulation**. We identify selective disclosure as a practically important and previously unformalized capability in LLM watermarking. To our knowledge, no prior work defines or addresses the problem of hierarchical payload verification with access control.
> 2. **Framework**. We propose the Hierarchical Vocabulary Routing framework, which enables verifiers at different authorization levels to recover different payload subsets while preserving sampling unbiasedness.
> 3. **Efficiency**. We contribute algorithmic improvements to the detection procedure and provide a low latency implementation (see our response to Q1 from Reviewer `BExV`).  Both are necessary to make watermark verification practical on a large scale of online content.
>
> In summary, although prior work uses multi-layer structures, we introduce a new problem setting and a fundamentally different mechanism to address it. We believe this clarification demonstrates the originality of our contribution and that it represents a meaningful advance beyond existing approaches.

---

> > ### Author Rebuttal · Reviewer_sHdd · 2026-04-03
> >
> > Please note that prior work can be readily extended to the multi-bit setting. For example, in SynthID, one can simply decompose the detector into $K$ independent components, where the detection function of the i-th bit is $g_i$.
> > Under this view: (a) multi-layer watermarking is not novel, and (b) the use of Gumbel-max sampling is also not novel. The authors’ primary contribution appears to be separating $g_i$ from the original detector and treating it as an independent detector for the i-th bit (and previous work [3] adapted somehow similar idea). In my opinion, this level of novelty does not meet the bar for ICML.

---

> > > ### Author Response · Authors · 2026-04-04
> > >
> > > We thank the reviewer for the quick follow-up.
> > >
> > > > Please note that prior work can be readily extended to the multi-bit setting. For example, in SynthID, one can simply decompose the detector into $K$ independent components, where the detection function of the i-th bit is $g_i$.
> > >
> > > This extension may be conceptually possible, but whether this specific proposal achieves competitive multi-bit performance in practice remains unclear without the supporting evidence. Analyzing this particular extension is outside the main scope of our work.
> > >
> > > > Under this view: (a) multi-layer watermarking is not novel, and (b) the use of Gumbel-max sampling is also not novel.
> > >
> > > (a) We respectfully point out that our method and prior multilayer reweighting approaches rely on two **structurally distinct** mechanisms, and therefore should be considered separately.
> > >
> > > 1. **Single sampling**. In methods [1] and [2], multiple layers of probability reweighting are applied over the **full vocabulary**, followed by a **single sampling event** to generate the token.
> > >
> > > 2. **Multiple sampling**. In HeRo, each token is generated each token through **$L$ sequential sampling stages**. At routing stage $\ell$, the sampler draws from a $K_\ell$​-way categorical distribution over aggregated chunk probabilities (meta-tokens), and the selected chunk defines the candidate set for stage $\ell+1$. This induces a nested structure
> > > $
> > > \mathcal{V} = C^{(0)} \supset C^{(1)} \supset \cdots \supset C^{(L)}=\\{x_t\\},
> > > $
> > > which does not exist in prior methods.
> > >
> > > We further note that prior multilayer reweighting approaches are **complementary** to HeRo: within each stage, one could further apply multilayer reweighting over the current distribution to strengthen the watermark signal.
> > >
> > > (b) We instantiate our framework using Gumbel-max sampling because it is simple and standard, but do not claim its use as a novelty.
> > >
> > > > The authors’ primary contribution appears to be separating $g_i$ from the original detector and treating it as an independent detector for the i-th bit (and previous work [3] adapted somehow similar idea).
> > >
> > > The $g_i$ decomposition is specific to SynthID, whose tournament sampling already contains multiple layers. By contrast, there is **no natural multilayer structure** in other zero-bit watermarking methods such as Green-Red List or Gumbel-Max watermark from which a corresponding $g_i$ can be naturally separated. Our understanding of [3] is different from the interpretation above. In [3], each token is assigned to a one message bit, and multilayer reweighting is used to strengthen that single bit embedding. This is not the same as embedding multi-bit information within a single token generation in the sense considered here.
> > >
> > >
> > > >  In my opinion, this level of novelty does not meet the bar for ICML.
> > >
> > > We have clarified the distinction between our framework and prior works above. We want to emphasize that our contribution is **not only** the proposed framework alone, but also includes (i) the problem formulation of selective disclosure, which to our knowledge has not been defined or addressed in prior LLM watermarking work, and (ii) algorithmic and implementation contributions to efficient detection.
> > >
> > > We hope these clarifications help resolve the concerns.

---

### Official Review · Reviewer_BExV · 2026-03-24

**Soundness:** 2
**Presentation:** 3
**Significance:** 2
**Originality:** 3
**Overall Recommendation:** 4
**Confidence:** 3

**Summary:**

This paper proposes HeRo , a watermarking framework for LLMs designed to achieve fine-grained access control over watermark data. Existing multi-bit watermarking methods typically follow an "all-or-nothing" disclosure model: once a verifier possesses the secret key, they can decrypt all embedded metadata. The core idea of HeRo is to guide the token sampling process through hierarchical vocabulary partitioning to embed watermarks in layers. Lower-level information is statistically hidden within the upper layers, remaining inaccessible to verifiers who lack the corresponding authorized keys. This method is statistically unbiased, achieving high detection precision and low latency while maintaining the original text quality.

**Compliance With Llm Reviewing Policy:**

Affirmed.

**Final Justification:**

Considering the authors' rebuttal and the other reviewers' comments, I maintain my score.

**Key Questions For Authors:**

1. As the number of layers increases, the vocabulary space allocated to each routing step shrinks exponentially. Such a constrained space could significantly compromise both watermark precision and security. Given that the experimental results already show a notable discrepancy in accuracy between the two existing layers, can this scheme realistically support a higher number of hierarchical divisions without degrading the quality of the generated text?
2. The decoding process relies on the evidence function $E_v$ . Would the progressive contraction of the vocabulary space—caused by increasing the number of layers—adversely affect the estimation accuracy of $E_v$, thereby imposing a practical limit on the maximum number of layers?
3. The statistical unbiasedness of HeRo appears to be intrinsically linked to the Gumbel-Max sampling method. As multi-layer embedding leads to a stepwise reduction in the available vocabulary space, does the underlying vocabulary for Gumbel-Max sampling change accordingly at each step? If it does change, would this shift compromise the theoretical unbiasedness? A more detailed and rigorous explanation regarding this mechanism in the context of Theorem 4.1 is necessary.
4. During the sampling process, the paper mentions using a "Mask" to handle cases where the token context is highly repetitive. Does this treatment significantly reduce the embedding density of effective information, especially within long-form texts containing frequent repetitive structures?

**Limitations:**

Yes.

**Strengths And Weaknesses:**

Strengths

1. This work proposes the idea of hierarchical embedding for LLM watermarking, effectively addressing the challenge of fine-grained access control. This holds significant practical value in the context of privacy protection and tiered regulation, demonstrating clear motivation and strong novelty.
2. The paper ingeniously models the watermarking process as a hierarchical decision-making task within the vocabulary space. This approach to multi-layer embedding is technically sound and supported by rigorous theoretical derivation.
3. The authors provide a strict mathematical proof of statistical unbiasedness, which offers crucial theoretical grounding for maintaining model generation quality (e.g., perplexity).
4. The experimental results cover multiple dimensions with an adequate set of baselines. Notably, HeRo achieves significant optimization in inference overhead and demonstrates superior performance across various metrics.

Weaknesses
1. As the number of layers increases, the vocabulary space allocated to each routing step shrinks exponentially. Such constrained space may negatively impact watermark precision and security. Since the paper only evaluates a two-layer depth, it remains unclear whether deeper layers would suffer from higher bit-error rates or diminished security, thereby hindering practical deployment. (Indeed, experimental data shows a notable gap in accuracy between the two layers).
2. The methodology appears to be an extension of single-bit watermarking to a multi-layer structure, still relying on the standard Gumbel-Max sampling. From an architectural standpoint, the innovation feels somewhat incremental. Furthermore, it is questionable whether this scheme can support more complex real-world scenarios beyond the two layers demonstrated.
3. The claimed unbiasedness of HeRo seems heavily dependent on Gumbel-Max sampling, yet this dependency is not explicitly reflected in the proof of Theorem 4.1. Additionally, as the vocabulary space progressively narrows across layers, does the corresponding Gumbel-Max sampling distribution need to be adjusted? If so, would these adjustments compromise the theoretical unbiasedness? A more comprehensive and detailed explanation is required.

---

> ### Author Rebuttal · Authors · 2026-03-31
>
> We thank the reviewer for the positive evaluation of the motivation, technical soundness, and empirical performance of our work. Below we clarify the raised concerns and provide additional evidence.
> ## Response to Q1: Hierarchy depth vs. generation quality
> We emphasize that increasing the number of hierarchical layers does not degrade generation quality. This follows from the statistical unbiasedness of our framework, which preserves the original sampling distribution.
>
> The table below reports perplexity under different hierarchical configurations, where Hero $^L(b_1,\cdots,b_L)$ denotes an $L$-level hierarchy in which the $l$-th layer discloses $b_l$-bits. Across all settings, the PPL remains very close to that of text generated without watermarking, indicating that increasing the number of hierarchical divisions does not degrade generation quality.
>
> |Perplexity|n=200|n=400|
> |-|-|-|
> |No watermark|4.31|4.16|
> |HeRo $^1$(8)|4.26|4.13|
> |HeRo $^2$(4,4)|4.31|4.15|
> |HeRo $^4$(2,2,2,2)|4.31|4.15|
> |HeRo $^8$(1,1,1,1,1,1,1,1)|4.30|4.13|
>
> We refer to our response to Q7 from Reviewer `Ytbb` for a discussion on detectability under deeper hierarchies.
> ## Response to Q2: Scalability of hierarchical depth (number of layers)
> By our watermark construction, the maximum number of layers is bounded by $\lfloor\log_2 V\rfloor$, where $V$ denotes the vocabulary size. For Llama2-7B ($V=32,000$), this corresponds to at most 14 layers. Importantly, this upper bound is not a practical limitation. In practice, only a small number of layers (e.g., 2–4) is sufficient to achieve fine-grained access control, which is the primary goal of our framework.
>
> Increasing the number of layers mainly affects how the total payload is distributed: deeper hierarchies allocate fewer bits per layer, which may reduce per-layer evidence. This explains the observed accuracy gap across layers, but does not impact generation quality (as shown in Q1). Therefore, our method is practically applicable in realistic multi-level scenarios without requiring deep hierarchies.
> ## Response to Q3: Dependence of unbiasedness on Gumbel-Max and changing sampling space
> We clarify that the unbiasedness of HeRo does not depend on Gumbel-Max specifically. Theorem 4.1 holds for any statistically unbiased sampling rule. Gumbel-Max is used in our implementation because it is simple and standard, but it is not essential for the theoretical guarantee. In other words, it can be replaced by other unbiased rules without affecting the theoretical guarantee.
>
> Regarding the reviewer’s question: the sampling object does change across stages (from tokens to “meta-tokens”), but this does not affect unbiasedness.
>
> **What is sampled at each stage.**
> At the routing stage $\ell$, we do not sample directly from individual tokens. Instead, the current candidate set $C^{(\ell-1)}$ is partitioned into $K_\ell$ disjoint chunks. Each chunk is treated as a “meta-token” with probability equal to its aggregated mass $\sum_{x \in C^{(\ell)}} P_t(x)$. We then apply Gumbel-Max over these $K_\ell$ meta-tokens. While the sampling object changes across stages, each step samples from the correct (coarsened) distribution.
>
> **Why unbiasedness is preserved.**
> The key point is that the hierarchical procedure samples according to consistent conditional distributions at every stage. For any token $v$, its probability under the hierarchical sampler can be decomposed into two parts:
>
> (1) The probability that the routing stages select the sequence of chunks leading to $v$. At stage $\ell$, conditioned on $C^{(\ell-1)}$, the selected chunk satisfies $\mathbb{P}(s_\ell=i_\ell(v)\mid C^{(\ell-1)})=\frac{\sum_{x\in C^{(\ell)}}P_t(x)}{\sum_{x\in C^{(\ell-1)}}P_t(x)}.$ Multiplying over $\ell=1,\dots,L-1$, this yields $\sum_{x\in C^{(L-1)}}P_t(x)$.
>
> (2) The probability of sampling $v$ within the final chunk $C^{(L-1)}$, which is $\frac{P_t(v)}{\sum_{x\in C^{(L-1)}}P_t(x)}.$ Multiplying these two terms, the chunk mass cancels exactly, giving $\mathbb{P}(x_t=v)=P_t(v).$
>
> Therefore, unbiasedness is preserved regardless of how the sampling space evolves across stages.
>
> We will revise the manuscript to make this argument clearer.
> ## Response to Q4: Reduced effective embedding density due to masking
> Repeated context masking does not significantly reduce the effective embedding density. Masking only skips positions that share the same h-gram context. Embedding at such positions would introduce redundant and correlated signals, which do not improve and may even hurt detection performance. Therefore, masking removes low-quality embedding positions rather than reducing useful signals. Empirically, we observe that detectability remains high under this strategy. Similar approaches have also been adopted in prior work [1,2], supporting its effectiveness in practice.
>
> [1] Unbiased watermark for large language models, ICLR 2024.
>
> [2] StealthInk: A Multi-bit and Stealthy Watermark for Large Language Models, ICML 2025.

---

> > ### Author Rebuttal · Reviewer_BExV · 2026-04-07
> >
> > Thank you for the additional experiments. Some of my concerns have been addressed. However, I note that Reviewer sHdd has raised concerns about insufficient novelty, and this issue remains unresolved. I maintain my score.

---

> > > ### Author Response · Authors · 2026-04-07
> > >
> > > Thank you for the follow-up. We would like to briefly summarize the novelty discussion with Reviewer `sHdd` for convenience.
> > >
> > > First, our contribution is not limited to proposing another watermarking framework. Beyond the framework itself, the paper also introduces the **problem formulation of selective disclosure** in LLM watermarking, a practically relevant capability largely missing from prior watermarking schemes, and also contributes an **efficient verification procedure** with lower latency. In this sense, the novelty of the work is broader than a single architectural design choice.
> > >
> > > Second, we clarify the distinction between HeRo and the multi-layer mechanisms mentioned by Reviewer `sHdd`. Prior multi-layer watermarking methods apply multi-layer reweighting over the full vocabulary, followed by a single sampling step, to strengthen a shared watermark signal. In contrast, HeRo targets a different objective of selective disclosure, and achieves it through a structurally different mechanism: **hierarchical vocabulary routing** through sequential sampling stages over aggregated chunk probabilities, progressively restricting the candidate set and encoding different portions of the payload to different authorization levels.
> > >
> > > In other words, the two lines of work differ in both **what they aim to support** (a single shared signal vs. layer-wise different payloads) and **the machanism they use** (single sampling with multi-layer reweighting vs. multi-stage sequential sampling with nested partitioning).
> > >
> > > We would like to note that Reviewer `sHdd` raised their score from 2 to 3 after the discussion. We hope the summary above helps address the novelty concern, and we would be glad to provide any further clarification if helpful.

---

### Decision · Program_Chairs · 2026-04-30

**Decision:**

Accept (regular)

**Comment:**

HeRo proposes a hierarchical vocabulary routing framework for multi-bit LLM watermarking. The reviews are leaning towards a positive evaluation, but some key unresolved concerns remain.

- Two of four reviewers (sHdd and BExV) question the novelty of the technical contribution. The highest-confidence reviewer (sHdd) identifies prior multi-layer watermarking work (SynthID) that is similar in the core mechanism. The authors' rebuttal depends on distinguishing "multi-stage sequential sampling" from "single sampling with multi-layer reweighting." The reviewer directly rebuts this by pointing out that SynthID's tournament sampling was originally conceived as multiple sampling and only simplified to single sampling for computational efficiency. The authors did not engage with this rebuttal substantively, instead moving to problem formulation and efficiency as independent contributions. This fallback is narrower than the original framing.
- Reviewer BExV independently described the methodology as "an extension of single-bit watermarking to a multi-layer structure" whose innovation "feels somewhat incremental."
- A main contribution is under-evaluated. The paper's central claim is support for multi-level access control, yet the main evaluation covers only a 2-layer hierarchy.
- Presentation obscures the contribution. Reviewer Ytbb notes that the notation is "overly complex," the level of abstraction "significantly hinders readability," and "the precise implementation details remain obscure." Two other reviewers also had difficulties in separating HeRo's contribution from related work.

Considering the reviews, the accept decision by CcxH was made with low confidence and did not consider technical details. Both weak accepts raise substantive concerns, and BExV's score is low due to the novelty concern. The weak reject is the highest-confidence review and has a defensible position in citing prior work.